# SIPO: STABILIZED AND IMPROVED PREFERENCE OPTIMIZATION FOR ALIGNING DIFFUSION MODELS

## ABSTRACT

Preference learning has garnered extensive attention as an effective technique for aligning diffusion models with human preferences in visual generation tasks. However, existing alignment approaches such as Diffusion-DPO suffer from two fundamental challenges: training instability caused by high gradient variances at various timesteps and high parameter sensitivities, and off-policy bias arising from the discrepancy between the optimization data and the policy model's distribution. Our first contribution is a systematical analysis of the diffusion trajectories across different timesteps and identify that the instability primarily originates from early timesteps with low importance weights. To address these issues, we propose SIPO, a Stabilized and Improved preference Optimization framework for aligning diffusion models with human preferences. Concretely, a key gradient, *i.e.,* DPO-C&M is introduced to facilitate stabilize training by clipping and masking uninformative timesteps. Followed by a timestep aware importance reweighting paradigm to fully correct off-policy bias and emphasize informative updates throughout the alignment process. Extensive experiments on various baseline models, including image generation models on SD1.5, SDXL, and video generation models CogVideoX-2B, CogVideoX-5B, and Wan2.1-1.3B, demonstrate that our SIPO consistently promotes stabilized training and outperforms existing alignment methods, with meticulous adjustments on parameters. Overall, these results highlight the importance of timestep-aware alignment and and provide valuable guidelines for improved preference optimization in diffusion models.

## 1 INTRODUCTION

Recent advances in diffusion models (Ho et al., 2020a; Jiaming Song, 2021) have revolutionized the field of generative models, facilitating remarkable experiences in generating high-quality images and videos (Esser et al., 2024; Gen-3, 2024; Tan et al., 2025; Kuaishou, 2024; hailuo, 2024; Wang et al., 2025a; Yang et al., 2024b; Kong et al., 2024). To further encourage diffusion models to produce user-preferred outputs, recent efforts incorporate alignment techniques, *e.g.,* Direct Preference Optimization (DPO) (Rafailov et al., 2023), to align models with human aesthetics and preferences. For instance, approaches including VADER(Prabhudesai et al., 2024), SPIN Diffusion (Yuan et al., 2024), DDPO (Black et al., 2023), D3PO (Yang et al., 2024a), and Diffusion DPO (Wallace et al., 2024) have achieved improved performance in producing user-preferred images.

Despite these growing interests, aligning diffusion models directly with DPO encounters two fundamental challenges. On one hand, the denoising process is characterized by a long-horizon with varied timesteps and stochastic trajectories throughout the training process. On the other hand, preferences used for optimization, usually derived from specific references models on collected out-of-distribution data, might be noisy and biased to the distribution of the optimized policy model. Consequently, the training dynamics might be unstable due to inaccurate off-policy score estimation. These challenges are further exacerbated in the realm of video generation, where maintaining temporal coherence and smooth motion is crucial. This context underscores the need for a deeper investigation into the dynamics of preference alignment under diffusion models, as well as developing effective strategies for stabilized and improved human alignment. Accordingly, we systematically analysis these challenges and identify the underlying causes.

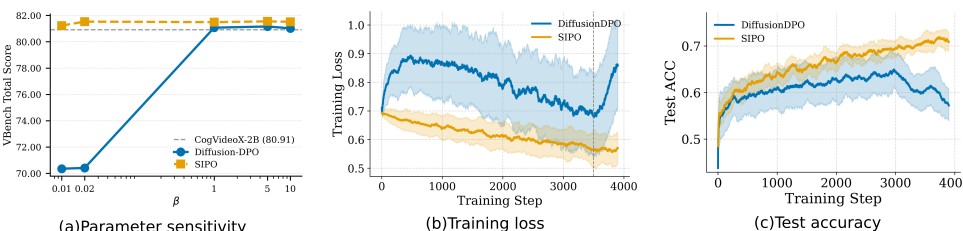

Figure 1: Comparison of SIPO and DiffusionDPO. (a) **Parameter sensitivity:** SIPO is more robust while DiffusionDPO fluctuates with $\beta$. (b) **Training loss:** DiffusionDPO shows unstable loss with late-stage increase, while SIPO decreases smoothly until convergence. (c) **Test accuracy:** DiffusionDPO degrades in later training, while SIPO steadily improves.

The training instability exhibits in several aspects. First, *the intense sensitivity on the parameter $\beta$* of DPO, which balances the discrepancy between the optimized model and the reference model. Fig. 1 (a) shows that larger $\beta$ values make updates more conservative whereas smaller ones lead to performance degradation due to aggressive updates. Following this, prior arts Wu et al. (2024); Wallace et al. (2024) indicate that increasing $\beta$ or employing a dynamic schedule can stabilize training. Second, *unstable loss dynamics* illustrated in Fig. 1 (b). Non-monotonic loss patterns reflect that the training loss can rebound and, in extreme cases, grow without bound, resulting in training collapse Park (2024). Third, the *timestep instability*. Previous works show that uniform timestep sampling underperforms emphasizing intermediate steps (Karras et al., 2022). Similarly, performing preference optimization at middle-to-late timesteps improves alignment performance as indicated in Fig. 1, while aligning earlier timesteps often decreases the performance (see Fig. 2).

The other key challenge is the off-policy mismatch. Specifically, labeled data for preference optimization is typically constructed from a fixed reference model (*e.g.,* a SFT multimodal Model (Wang et al., 2025b)) rather than originating from the optimized policy model. unfortunately, this forms a distributional gap where the training data captures a narrow subset of the state-action space, whereas the model is required to learn precise alignment across a significantly broader space (Ren & Sutherland, 2025). That is, the model is compelled to generalize beyond the observed data, leading to potentially leading to uncontrolled training dynamics, degraded generalization, and even collapse (Pal et al., 2024; Feng et al., 2024). Additionally, empirical studies (see Fig. 3) indicate that DPO might degrade the synthesis quality over extended training period. Such degradation is attribute to unsatisfactory overfitting and reward hacking, where training reward increases but the evaluation performance decreases. This limitation is further exacerbated in offline optimization that relies on fixed data Park et al. (2024); Chen et al. (2024b); Azar et al. (2024) without on-policy exploration Xiong et al. (2023); Liu et al. (2023b), limiting adaptation to underrepresented regions of training data.

To address the aforementioned challenges, this paper proposes a stabilized and improved preference (**SIPO**) framework for aligning diffusion models. Through a comprehensive analysis on the training dynamics of different timesteps, we identify that preference optimization at earlier timesteps causes instability and more focus should be concentrated on middle-to-late timesteps. Accordingly, we innovatively develop *DPO-C&M* to addresses training instability caused by mismatched or noisy samples by applying timestep-aware masking and gradient clipping, thus mitigating timestep-dependent variance. Furthermore, a *timestep-wise clipped importance re-weighting* scheme is incorporated to effectively and adaptively reduce off-policy bias. Together, our proposed framework contributes to stabilize the training dynamics and facilitate the alignment process. To testify the effectiveness of our proposed SIPO, we conduct extensive experiments on aligning different models with human preferences across various baseline models, in both image and video generation settings. The results consistently reflect that SIPO significantly stabilizes the training process and outperforms comparison approaches, *e.g.,* Diffusion-DPO, with more robust parameter sensitivity(as shown in Fig. 1).

To sum up, this paper makes three-fold contributions: 1) We conduct a systematic analysis of diffusion-based preference optimization and find that early timesteps tend to destabilize DPO training, whereas the substantial gains are contributed by mid-to-late timesteps. 2) We propose SIPO, a principled importance-weighted optimization framework that stabilizes training and mitigates off-policy mismatch by reweighting gradients according to sample likelihood and timestep quality. 3) We validate that SIPO effectively stabilize the preference optimization process and consistently outperforms existing alternatives on both image and video generation tasks across various baseline models, in both automatic and human evaluations.

## 2 BACKGROUND

**Diffusion Models.** Diffusion models (Ho et al., 2020b; Nichol & Dhariwal, 2021) define a forward noising process and a learned reverse process. Given a clean sample $x_0 \sim q(x_0)$, the forward process gradually perturbs it:

$$q(x_t \mid x_{t-1}) = \mathcal{N}\left(x_t;\ \sqrt{1 - \beta_t}\, x_{t-1},\ \beta_t \mathbf{I}\right), \tag{1}$$

where $x_t \in \mathbb{R}^d$ is the latent at step $t$, and $\{\beta_t\}_{t=1}^T$ is a variance schedule. The reverse process is parameterized by a neural network:

$$p_\theta(x_{t-1} \mid x_t) = \mathcal{N}(x_{t-1}; \mu_\theta(x_t, t), \Sigma_\theta(x_t, t)). \tag{2}$$

In practice, the model is trained to predict the injected noise $\epsilon \sim \mathcal{N}(0, \mathbf{I})$ via

$$\mathcal{L}_{\mathrm{DM}}(\theta) = \mathbb{E}_{t, x_0, \epsilon}\left[\, \|\epsilon - \epsilon_\theta(x_t, t)\|^2 \,\right], \tag{3}$$

where $x_t = \sqrt{\bar{\alpha}_t} x_0 + \sqrt{1 - \bar{\alpha}_t}\, \epsilon$ and $\bar{\alpha}_t = \prod_{s=1}^t (1 - \beta_s)$.

**Importance Sampling.** Importance sampling (Owen, 2013) estimates expectations under a target distribution $p(x)$ using samples from a proposal $q(x)$:

$$\mathbb{E}_p[f(x)] = \int f(x)p(x)dx = \mathbb{E}_q\left[f(x)\frac{p(x)}{q(x)}\right], \tag{4}$$

where $\frac{p(x)}{q(x)}$ is the importance weight. It is widely used in reinforcement learning, e.g., PPO (Schulman et al., 2017) and GRPO (Shao et al., 2024), to correct distribution mismatch while stabilizing updates by clipping extreme weights.

In diffusion models, importance sampling can be applied by comparing the learned reverse process with either the forward posterior or a previous model iteration:

$$w(t) = p_\theta(x_{t-1} \mid x_t)/q(x_{t-1} \mid x_t, x_0) \quad \text{or} \quad w(t) = p_\theta(x_{t-1} \mid x_t)/p_{\mathrm{old}}(x_{t-1} \mid x_t). \tag{5}$$

We follow DDPO (Black et al., 2023) to compute transition densities under the standard Gaussian diffusion framework, ensuring consistency and enabling efficient importance weighting.

**DPO Objective.** RLHF (Reinforcement Learning from Human Feedback) (Ouyang et al., 2022) can be viewed as being grounded in two core objectives: the first focuses on reward model learning, and the second on policy optimization through reward maximization.

The reward model is trained using pairwise preference data via the binary logistic loss:

$$\mathcal{L}_{\mathrm{reward}} = -\mathbb{E}_{\mathbf{c}, \mathbf{x}_0^w, \mathbf{x}_0^l}\left[\log \sigma\left(r(\mathbf{c}, \mathbf{x}_0^w) - r(\mathbf{c}, \mathbf{x}_0^l)\right)\right]. \tag{6}$$

The policy is then optimized to maximize expected reward while staying close to a reference distribution:

$$\max_{p_\theta} \mathbb{E}_{\mathbf{c} \sim \mathcal{D}_{\mathbf{c}},\, \mathbf{x}_0 \sim p_\theta(\mathbf{x}_0 | \mathbf{c})} \left[r(\mathbf{c}, \mathbf{x}_0)\right] - \beta\, \mathbb{D}_{\mathrm{KL}}\left[p_\theta(\mathbf{x}_0 | \mathbf{c}) \,\|\, p_{\mathrm{ref}}(\mathbf{x}_0 | \mathbf{c})\right]. \tag{7}$$

To derive the training objective of Direct Preference Optimization (DPO), the reward function is reformulated based on the optimization target in Eq. 7. Specifically, the reward can be written as

$$r(c, x_0) = \beta \log \frac{p_\theta^*(x_0 \mid c)}{p_{\mathrm{ref}}(x_0 \mid c)} + \beta \log Z(c), \tag{8}$$

where $Z(c) = \sum_{x_0} p_{\mathrm{ref}}(x_0 \mid c) \exp\left(\frac{r(c, x_0)}{\beta}\right)$. By further substituting this form into the objective (Eq. 6), the final DPO optimization criterion is obtained:

$$\mathcal{L}_{\mathrm{DPO}}(\theta) = -\mathbb{E}_{c,\, x_0^w, x_0^\ell}\left[\log \sigma\left(\beta \log \frac{p_\theta(x_0^w \mid c)}{p_{\mathrm{ref}}(x_0^w \mid c)} - \beta \log \frac{p_\theta(x_0^\ell \mid c)}{p_{\mathrm{ref}}(x_0^\ell \mid c)}\right)\right]. \tag{9}$$

**Diffusion-DPO.** Prior extensions, including DDPO (Black et al., 2023), D3PO (Yang et al., 2024a), and SPIN Diffusion (Yuan et al., 2024), highlight the promise of preference learning but often incur

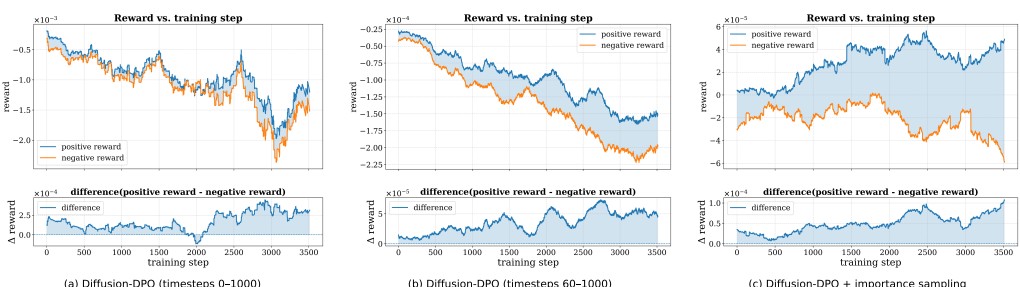

(a) Diffusion-DPO (timesteps 0–1000)  (b) Diffusion-DPO (timesteps 60–1000)  (c) Diffusion-DPO + importance sampling

Figure 2: Reward dynamics in Diffusion-DPO under different training settings. (a) Random timesteps (0–1000) give unstable rewards and a small gap. (b) Restricting to 60–1000 stabilizes training but reduces rewards, while enlarging the gap. (c) Dynamic timestep clipping with importance sampling further enlarges the gap, with positive rewards increasing and negative rewards decreasing, better matching the DPO objective.

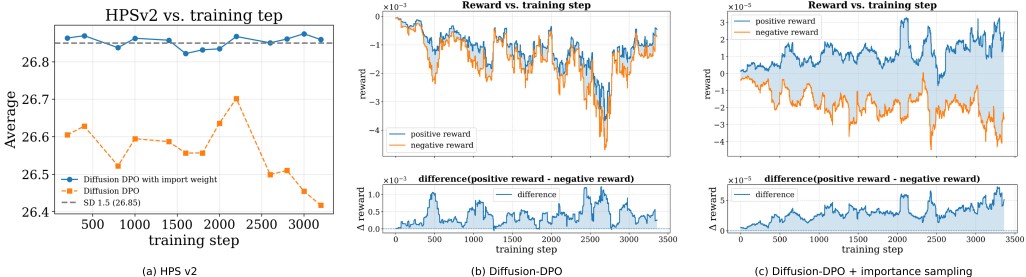

(a) HPS v2  (b) Diffusion-DPO  (c) Diffusion-DPO + importance sampling

Figure 3: Effects of training on *unlike* datasets(training on data the model cannot generate). (a) HPSv2 accuracy: vanilla Diffusion-DPO causes a performance drop, whereas adding importance sampling preserves the original model accuracy (no degradation). (b) Vanilla training causes oscillations and an unstable small gap. (c) With importance sampling, training better matches the DPO objective: positive rewards rise, negatives fall, enlarging the gap and stabilizing learning.

high computational costs. Diffusion-DPO (Wallace et al., 2024) overcomes these limitations by reformulating DPO for diffusion and introducing a more tractable surrogate objective.

To make training efficient, the objective is approximated by comparing single-step reverse transitions at a randomly sampled time step $t \in \{1, \dots, T\}$:

$$L(\theta) = -\mathbb{E}_{t,x_t^{\mathrm{w}},x_t^{\mathrm{l}}} \left[ \log \sigma \left( -\beta T \omega(\lambda_t) \cdot \Delta\ell(\theta) \right) \right], \tag{10}$$

where $\sigma(\cdot)$ denotes the sigmoid function, $\omega(\lambda_t)$ is a timestep-dependent weighting function, and $\Delta\ell(\theta)$ measures the model's preference alignment error based on noise prediction:

$$\Delta\ell(\theta) = \|\epsilon^{\mathrm{w}} - \epsilon_\theta(x_t^{\mathrm{w}}, t)\|^2 - \|\epsilon^{\mathrm{w}} - \epsilon_{\mathrm{ref}}(x_t^{\mathrm{w}}, t)\|^2 - \left( \|\epsilon^{\mathrm{l}} - \epsilon_\theta(x_t^{\mathrm{l}}, t)\|^2 - \|\epsilon^{\mathrm{l}} - \epsilon_{\mathrm{ref}}(x_t^{\mathrm{l}}, t)\|^2 \right). \tag{11}$$

Here, $\epsilon^{\mathrm{w}}, \epsilon^{\mathrm{l}} \sim \mathcal{N}(0, I)$ are noise samples used to construct $x_t$ via the forward process.

## 3 METHOD

### 3.1 ANALYSIS ON PREFERENCE LEARNING AND MOTIVATIONS

Motivated by these considerations, we analyze preference signals and training stability along the diffusion trajectory using Stable Diffusion 1.5 (Rombach et al., 2021) on the PickaPic dataset (Kirstain et al., 2023), computing rewards via Eq. 8 and examining whether DPO training consistently increases positive rewards, suppresses negative rewards, and enlarges their gap.

**Importance Weights Reveal Stability.** Importance sampling allows unbiased estimation when training distributions differ from the target distribution by reweighting samples according to their likelihood ratio, thus improving efficiency and stability in learning (Owen, 2013). However, excessively small importance weights can lead to high variance and ineffective gradient updates, reducing

training stability and slowing convergence (Schulman et al., 2017; Mnih & Rezende, 2016; Graves, 2011). To investigate the impact of importance sampling on the stability of Diffusion-DPO, we incorporate the importance weight $w(t)$ (Eq. 5) into training and dynamically prune timesteps with weights below 0.9. As shown in Fig. 2c, this strategy improves alignment with the DPO optimization objective: positive-sample rewards steadily increase while negative-sample rewards are suppressed. In contrast, the original Diffusion-DPO training (Fig. 2 (a)) exhibits large fluctuations, with both positive and negative rewards decreasing over time. Consequently, model performance deteriorates as training progresses, mirroring the trend in Fig. 1, where test accuracy first improves and then declines, based on experiments with the CogVideo video generation model.

**Effective Timesteps.** Building on the above analysis, we observe that discarding timesteps with low importance weights improves the stability of Diffusion-DPO training. As shown in Fig. 4, these low-weight regions correspond to the early timesteps: specifically, importance weights below 0.9 lie within the range of timesteps 0–63. This suggests that early timesteps negatively impact the stability of DPO training. To further validate this, we restrict training to the range of timesteps 60–1000 (Fig. 2 (b)). Compared to the original Diffusion-DPO training (Fig. 2 (a)), this setting improves the separation between positive and negative samples, but both rewards still decrease. Since importance weights evolve with

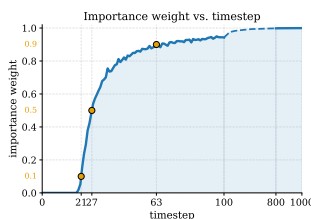

Figure 4: Importance weight $w(t)$ increases with timestep.

model updates, we further apply importance sampling (Fig. 2 (c)), yielding a more stable reward trajectory and better alignment with the DPO objective.

**Off-Policy Instability and Importance Weights.** Diffusion-DPO, like other offline preference optimization methods, suffers from a mismatch between the training distribution and the current policy, which shifts as the model is updated. To study this effect in diffusion models, we adopt Stable Diffusion 1.5 (Rombach et al., 2021) as the base model and train on the Rapidata human preference dataset (Rapidata, 2024), containing 700k preference-labeled images generated by stronger models such as Flux (Labs, 2024), and DALL·E 3 (OpenAI, 2023). Since SD1.5 can hardly reproduce this dataset, we term it an *unlike dataset*, which introduces a large distribution gap. On HPSv2 (Wu et al., 2023), vanilla Diffusion-DPO trained on this dataset exhibits performance degradation and unstable reward dynamics (Fig. 3 (a)(b)). Incorporating importance sampling mitigates this instability and preserves accuracy, though without clear improvements (Fig. 3 (a)(c)). Importantly, such instability is not limited to extreme mismatches: even with closer datasets, distributional drift naturally accumulates as training progresses, leading to performance decline in later stages (Fig. 1).

Our analysis shows that Diffusion-DPO instability arises from two factors: (i) low-importance early timesteps that produce noisy gradients, and (ii) distributional mismatch between training data and the current policy, causing off-policy drift and accuracy loss. Importance weights offer a principled way to guide training in a more reliable optimization regime.

### 3.2 IMPORTANCE-SAMPLED DIRECT PREFERENCE OPTIMIZATION

Motivated by these findings, we propose two complementary improvements. **DPO-C&M** stabilizes training by masking unreliable early-timestep updates and clipping gradients with importance weights as thresholds. Building on this, **SIPO** further corrects off-policy bias through timestep-wise clipped and re-weighted importance sampling, yielding more stable optimization.

**DPO-C&M** The core idea is to adjust each training sample's contribution based on how well it matches the reverse process. We introduce an importance weight $w(t)$, defined in Eq. 5, which compares the reverse model transition $p_\theta(x_{t-1} \mid x_t)$ with the forward noising process $q(x_t \mid x_0)$. To prevent high variance and instability, this weight is clipped within a fixed range:

$$\tilde{w}(t) = \text{clip}(w(t), 1 - \epsilon, 1 + \epsilon). \tag{12}$$

The clipped importance weight $\tilde{w}(t)$ serves two purposes. First, it rescales gradient updates to reflect the reliability of each sample. Second, it acts as a soft mask to suppress gradient flow from noisy regions where the forward and reverse paths diverge significantly. Combining this masked weighting mechanism with the original preference-based objective yields the DPO-C&M loss:

$$L_{\text{DPO-C\&M}}(\theta) = -\mathbb{E}_{\substack{(x_0^w, x_0^l) \sim \mathcal{D}, \, t \sim \mathcal{U}(0,T) \\ x_t^w \sim q(\cdot|x_0^w), \, x_t^l \sim q(\cdot|x_0^l)}} \tilde{w}(t) \cdot \log \sigma \left( -\beta T \omega(\lambda_t) \cdot \Delta\ell(\theta) \right), \tag{13}$$

where $\Delta\ell(\theta)$ is defined in Eq. 10. By selectively updating only well-aligned regions, DPO-C&M directly addresses the challenge of distribution mismatch of diffusion preference optimization.

**The proposed SIPO.** While DPO-C&M simply clips low-weight timesteps using importance weights as thresholds, it does not resolve the off-policy mismatch. To address this, we propose **SIPO** (Stabilized and Improved Preference Optimization), which introduces learnable importance weights to adaptively reweight updates, correct distribution shift, and further stabilize training.

Building on the standard preference optimization objective, we reinterpret it via importance sampling with adaptive reweighting, combining Eq. 7 and Eq. 12 to recast the original expectation over model samples into an importance-weighted expectation over off-policy data:

$$\max_{p_\theta} \ \mathbb{E}_{\mathbf{c}\sim\mathcal{D}_\mathbf{c},\, \mathbf{x}_0\sim q(\mathbf{x}_0|\mathbf{c})} \left[w_\theta \cdot r(\mathbf{c}, \mathbf{x}_0)\right] - \beta\, \mathbb{D}_{\mathrm{KL}} \left[p_\theta(\mathbf{x}_0|\mathbf{c}) \,\|\, p_{\mathrm{ref}}(\mathbf{x}_0|\mathbf{c})\right]. \tag{14}$$

Here, the importance weight $w_\theta = \frac{p_\theta(x_0|c)}{q(x_0|c)}$ corrects for the mismatch between the model distribution and the off-policy sampling distribution $q$, allowing optimization to be performed over pre-collected data. Moreover, the KL term acts as a regularizer that constrains the learned model from drifting too far away from the reference model $p_{\mathrm{ref}}$. Similar to GRPO (Shao et al., 2024), this regularization can be regarded as distribution-agnostic, since it depends only on the divergence between the current and reference models rather than the training data distribution. TIS-DPO (Liu et al., 2024) shows that this importance-sampling formulation yields an unbiased estimator for off-policy optimization.

To better understand how this objective guides the model distribution during training, we further reformulate it into a KL divergence form. This transformation makes the optimization direction explicit: it reveals that the model is implicitly learning to match a shaped target distribution $p^*$ that balances reward feedback with prior preferences. The reformulation is achieved by expressing the weighted reward term as a log-density ratio, leading to an equivalent form detailed in Appendix B.

$$\min_{p_\theta} \ \mathbb{E}_{\mathbf{c}\sim\mathcal{D}_\mathbf{c},\, \mathbf{x}_0\sim q(\mathbf{x}_0|\mathbf{c})} \left[\log\left(\frac{p_\theta(\mathbf{x}_0|\mathbf{c})}{p^*(\mathbf{x}_0|\mathbf{c})}\right) - \log Z(\mathbf{c})\right], \tag{15}$$

where $p^*(\mathbf{x}_0 \mid \mathbf{c})$ is the target distribution:

$$p^*(\mathbf{x}_0 \mid \mathbf{c}) = 1/Z(\mathbf{c}) \cdot p_{\mathrm{ref}}(\mathbf{x}_0 \mid \mathbf{c}) \cdot \exp\left(w_\theta/\beta \cdot r(\mathbf{c}, \mathbf{x}_0)\right). \tag{16}$$

The normalization constant is $Z(\mathbf{c}) = \sum_{\mathbf{x}_0} p_{\mathrm{ref}}(\mathbf{x}_0 \mid \mathbf{c}) \exp\left(\frac{1+\epsilon}{\beta} \cdot r(\mathbf{c}, \mathbf{x}_0)\right)$. This change of form makes explicit that SIPO minimizes the KL divergence between the current model and a shaped target distribution, effectively guiding $p_\theta$ toward reward-aligned behavior under off-policy sampling. This KL-based perspective shows that optimization aligns $p_\theta$ with $p^*$, making SIPO a direct learning procedure for the target distribution; rearranging $p^*$ yields the reward:

$$r(\mathbf{c}, \mathbf{x}_0) = \frac{\beta}{w_\theta} \cdot \left[\log\left(\frac{p^*(\mathbf{x}_0|\mathbf{c})}{p_{\mathrm{ref}}(\mathbf{x}_0|\mathbf{c})}\right) + \log Z(\mathbf{c})\right], \tag{17}$$

We substitute this reward into the pairwise logistic loss (Eq. 6) following the DPO framework (Rafailov et al., 2023), and apply it at each denoising step $t$, consistent with Diffusion-DPO (Wallace et al., 2024). This results in the final SIPO objective:

$$L_{\mathrm{SIPO}}(\theta) = -\mathbb{E}_{(x_0^\mathrm{w}, x_0^\mathrm{l})\sim\mathcal{D},\, t\sim\mathcal{U}(1,T)} \log \sigma\left(\tilde{w}_\theta(t) \cdot \psi(x_{t-1}^\mathrm{w} \mid x_t^\mathrm{w}) - \tilde{w}_\theta(t) \cdot \psi(x_{t-1}^\mathrm{l} \mid x_t^\mathrm{l})\right), \tag{18}$$

where $\psi(x_{t-1} \mid x_t) = \beta \cdot \log\left(\frac{p_\theta^*(x_{t-1}|x_t)}{p_{\mathrm{ref}}(x_{t-1}|x_t)}\right)$, and $\tilde{w}_\theta(t)$ is the clipped step-wise importance weight.

The step-wise importance weight is defined via inverse weighting, and for preference pairs, the maximum of the two is used:

$$\tilde{w}_\theta(x_t, t) = \mathrm{clip}\left(1/w_\theta(x_t, t),\, 1-\epsilon,\, 1+\epsilon\right), \quad \tilde{w}_\theta(t) = \max\left(\tilde{w}_\theta(x_t^\mathrm{w}, t),\, \tilde{w}_\theta(x_t^\mathrm{l}, t)\right). \tag{19}$$

The weighted version of DPO scales each log-ratio term by $q/p$, which adaptively adjusts step sizes by reinforcing underestimated preferred samples and downweighting overconfident ones. This imposes a soft upper bound near the reference distribution that mitigates over-optimization and reward hacking. Reweighting also improves robustness to distribution shift by emphasizing rare or difficult samples, while the finite peak of each term reduces sensitivity to $\beta$, yielding more stable and easier-to-tune training.

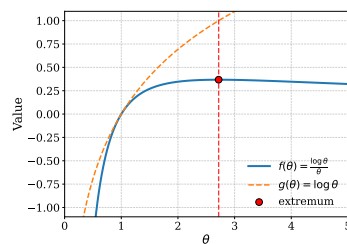

Figure 5: Log function and the weighted version.

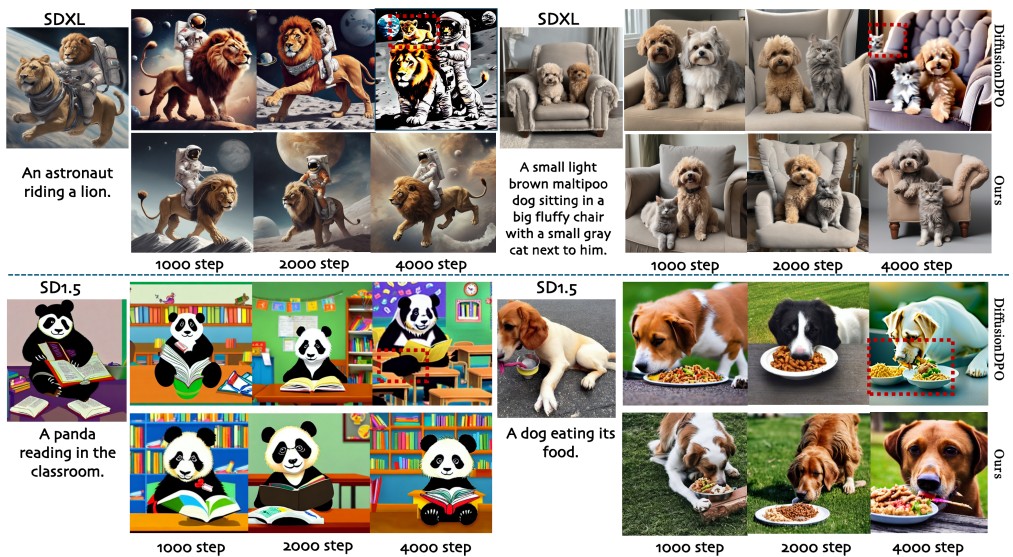

Figure 6: Visual comparison between Diffusion-DPO and our SIPO across different training steps.

# 4 EMPIRICAL RESULTS

## 4.1 IMPLEMENTATION DETAILS

We evaluate our proposed SDPO method on both text-to-video and text-to-image generation tasks. For video generation, we build upon CogVideoX-2B, CogVideoX-5B Yang et al. (2024b), and Wan-1.3B Wan et al. (2025), collecting 10k high-quality human preference pairs for training, and assess performance with VBench Huang et al. (2024), reporting aggregated Total, Quality, and Semantic scores. For image generation, we experiment with Stable Diffusion 1.5 and SDXL, training on the Pick-a-Pic dataset Kirstain et al. (2023), and evaluate using PickScore, ImageReward, HPSv2, and Aesthetic scores to capture both alignment and perceptual quality. Training configurations and hyperparameters are detailed in the Appendix D.

## 4.2 MAIN RESULTS

**Quantitative Results on Text-to-Video Generation.** We present quantitative results on text-to-video generation. As shown in Tab. 1, both DPO-C&M and SIPO consistently outperform Diffusion-DPO, with SIPO achieving the best overall performance. On CogVideoX-2B at 500 steps, Diffusion-DPO yields a modest gain (81.16 vs. 80.91 for the pretrained baseline), while DPO-C&M and SIPO further improve to 81.37 and 81.53. At 1000 steps, Diffusion-DPO collapses (67.28), indicating severe degradation in generation quality, whereas DPO-C&M and SIPO remain stable (81.46 and 81.68), demonstrating stronger robustness. To assess generality, we further evaluate on CogVideoX-5B and WanX-1.3B. SIPO achieves 82.28 and 84.78, respectively, significantly surpassing Diffusion-DPO, and showing that our approach also generalizes well to flow-based diffusion models such as WanX.

**Quantitative Results on Text-to-Image Generation.** A comparison of different post-training methods on text-to-image generation is conducted, reporting results on HPS, Aesthetic, ImageReward, and PickScore, along with their average. As shown in Tab. 2, SIPO consistently achieves the best overall performance with an average score of 7.4150. Specifically, SIPO surpasses Diffusion-DPO and DPO-C&M across multiple metrics, obtaining higher HPS (0.2763) and PickScore (22.01), while also maintaining competitive performance on Aesthetic (6.2451) and ImageReward (1.1257). Although SPIN-Diffusion achieves a slightly higher Aesthetic score, its overall performance is weaker than SIPO. These results demonstrate that SIPO provides a more balanced improvement across different evaluation metrics, leading to superior alignment with human preference in text-to-image generation. All evaluations are conducted on the Pick-a-Pic test dataset.

**Visualization Comparison.** As shown in Fig. 6, DiffusionDPO exhibits degradation under long training (e.g., 4000 steps), where consistency, aesthetics, and color fidelity decline, often producing

Table 1: Comparison results of different alignment methods on VBench.

| Model | Method | Total ↑ | Quality ↑ | Semantic ↑ |
|---|---|---|---|---|
| CogVideoX-2B @ 500 Steps | Pretrained | 80.91 | 82.18 | 75.83 |
| | + Diffusion-DPO | 81.16 | 82.32 | 76.49 |
| | + C&M | 81.37 | 82.55 | 76.69 |
| | + SIPO (ours) | **81.53** | **82.74** | **76.71** |
| CogVideoX-2B @ 1000 Steps | Pretrained | 80.91 | 82.18 | 75.83 |
| | + Diffusion-DPO | 67.28 | 68.37 | 62.93 |
| | + C&M | 81.46 | 82.65 | 76.72 |
| | + SIPO (ours) | **81.68** | **82.92** | **76.76** |
| CogVideoX-5B | Pretrained | 81.91 | 83.05 | 77.33 |
| | + Diffusion-DPO | 82.02 | 83.15 | 77.50 |
| | + C&M | 82.17 | 83.26 | 77.81 |
| | + SIPO (ours) | **82.28** | **83.37** | **77.91** |
| WanX-1.3B | Pretrained | 84.26 | 85.30 | 80.09 |
| | + Diffusion-DPO | 84.41 | 85.32 | 80.44 |
| | + C&M | 84.54 | 85.51 | 80.67 |
| | + SIPO (ours) | **84.78** | **85.73** | **81.02** |

Table 2: Comparison of different post-training methods on text-to-image generation.

| Model | HPS ↑ | Aesthetic ↑ | ImageReward ↑ | PickScore ↑ | Average ↑ |
|---|---|---|---|---|---|
| SD-1.5 | 0.2699 | 5.7691 | 0.8159 | 21.1983 | 7.0133 |
| SFT | 0.2749 | 5.9451 | 1.1051 | 21.4542 | 7.1948 |
| Diffusion-DPO | 0.2753 | 5.8918 | 1.0495 | 21.8866 | 7.2758 |
| SPIN-Diffusion | 0.2759 | **6.2481** | 1.1239 | 22.0024 | 7.4126 |
| DPO-C&M | 0.2749 | 6.1397 | 1.1753 | 21.8841 | 7.3683 |
| SIPO (ours) | **0.2763** | 6.2451 | **1.1257** | **22.0130** | **7.4150** |

halo artifacts. In contrast, SIPO maintains stable improvement across training, preserving semantic coherence and aesthetic appeal. Even at early timesteps, our method achieves competitive or superior visual results, highlighting its robustness and better alignment with human-preferred qualities.

**Human Evaluation.** A human evaluation is conducted on 200 prompts across diverse categories, where annotators rank videos generated by CogVideoX-2B, Diffusion-DPO, DPO-C&M, and SIPO along three dimensions. Results aggregated over multiple annotators (see supplementary) are shown in Fig. 7. SIPO achieves the highest share of first-place rankings (67%), clearly outperforming Diffusion-DPO and CogVideoX-2B, which are mostly ranked last. DPO-C&M performs slightly

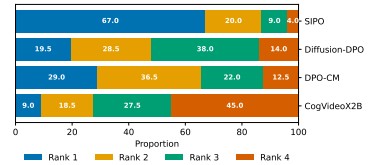

Figure 7: Human evaluation results.

worse than SIPO but still ranks higher than Diffusion-DPO, indicating that clipping and masking effectively reduce the impact of noisy timesteps.

### 4.3 ABLATION STUDY AND TRAINING DYNAMICS

**Hyperparameter $\beta$ Sensitivity Analysis** The temperature hyperparameter $\beta$, defined in Eq. (7), controls the strength of the KL-divergence penalty that prevents the optimized policy $\pi_\theta$ from deviating too far from the reference model $\pi_{\text{ref}}$. We evaluate both Diffusion-DPO and SIPO on the SD-1.5 model under a range of $\beta$ values, measuring performance on the HPSv2 benchmark after a fixed training duration. As shown in Fig. 8c, Diffusion-DPO is highly sensitive to $\beta$: at very small values its performance drops far below the pretrained baseline, it peaks around $\beta = 1$, and then slightly declines at $\beta = 10$. In contrast, SIPO exhibits only minor variation across the same range, consistently outperforming the baseline and achieving strong results even at $\beta = 0.02$. This demonstrates the robustness of SIPO and confirms that the importance-weighting mechanism effectively regularizes the model.

**Training Stability.** To examine training stability, we track the performance of SIPO and Diffusion-DPO during the training of the SD-1.5 model. Figure 8 reports results on both the HPSv2 bench-

Table 3: Evaluation results across different tasks. Higher is better.

| Model | Sin Obj. ↑ | Two Obj. ↑ | Counting ↑ | Colors ↑ | Position ↑ | Color Attri. ↑ | Overall ↑ |
|---|---|---|---|---|---|---|---|
| FLUX.1-dev | 0.98 | 0.93 | 0.75 | 0.93 | 0.68 | 0.65 | 0.82 |
| SFT | 0.96 | 0.95 | 0.77 | 0.94 | 0.66 | 0.67 | 0.83 |
| D3PO | 0.95 | 0.94 | 0.76 | 0.95 | 0.66 | 0.65 | 0.81 |
| Diffusion-DPO | 0.97 | 0.95 | 0.76 | 0.93 | 0.67 | 0.66 | 0.82 |
| SIPO (ours) | **0.98** | **0.95** | **0.79** | **0.95** | **0.73** | **0.69** | **0.85** |

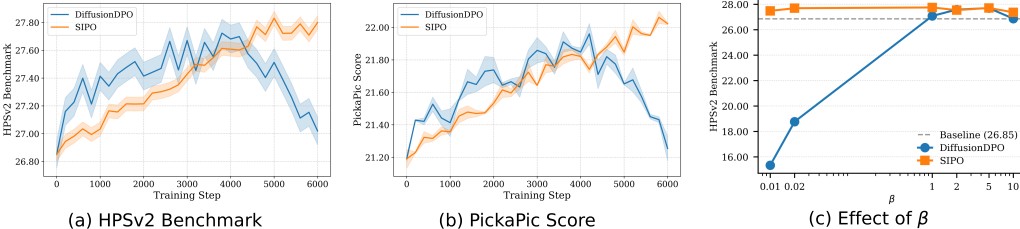

(a) HPSv2 Benchmark     (b) PickaPic Score     (c) Effect of $\beta$

Figure 8: Comparison of DiffusionDPO and SIPO. (a) On the HPSv2 benchmark, DiffusionDPO shows performance degradation in later stages, while SIPO remains stable. (b) On the PickaPic test, DiffusionDPO again declines while SIPO continues to improve. (c) Effect of $\beta$: DiffusionDPO is highly sensitive to small $\beta$, showing severe degradation, whereas SIPO is more robust.

mark(Fig. 8a) and the Pick-a-Pic score.(Fig 8b). Diffusion-DPO exhibits large fluctuations and eventually suffers a noticeable drop in performance, especially after 4000 training steps. In contrast, SIPO demonstrates smoother and more stable progress throughout training, consistently maintaining superior results at later stages. These findings highlight the robustness of SIPO against instability issues that arise during long training trajectories.

### 4.4 LEARNING FROM AI FEEDBACK.

**Preference Learning with AI Feedback.** Recent advances show that diffusion models can benefit from AI feedback, where generated outputs are automatically compared and ranked using pretrained scoring networks or verifiable reward models. Building on this idea, we evaluate SIPO against D3PO and Diffusion-DPO on the Geneval benchmark. Prompts are constructed by randomly composing objects, attributes, and spatial relations, following the Geneval protocol. To avoid overlap with models used in Geneval testing, we adopt YOLO-World as the reward model to detect objects and extract attributes such as count, position, and color, which are then scored against the input text following T2I-R1 (Jiang et al., 2025). This procedure yields 5,000 preference pairs for training, and high-scoring samples are further used for SFT. Different from SD-1.5, we use FLUX-dev as the base model, which is trained with flow matching and provides stronger generative capacity, further validating the applicability of our method to flow matching models.

## 5 CONCLUSIONS

In this work, we introduce SIPO, towards Stabilized and Improved Preference Optimization for aligning diffusion models with human preferences. We first perform a systematic analysis to probe the training dynamics across different timesteps, revealing that preference signals are more informative in middle-to-late timesteps. Based on this observation, we propose DPO-C&M to clip and mask less informative preference signals and incorporate a timestep-aware reweighting scheme to further facilitate the alignment process. Extensive results on various baseline models across both image and video generation tasks consistently reflect the superioty of our proposed method compared with existing alternatives. Moreover, SIPO remains robust under online and iterative training, offering a scalable and principled approach to aligning diffusion-based generation with human preferences.

## ETHICS STATEMENT

This work is based solely on publicly available datasets and widely used generative models, such as Stable Diffusion and CogVideoX. All preference data is either collected from open benchmarks (e.g., Pick-a-Pic, Rapidata) or generated automatically, without involving private or personally identifiable information. No human subjects are involved in ways that raise ethical concerns. The purpose of this work is to improve the stability and alignment of diffusion-based generative models. Therefore, we believe that this research does not pose ethical risks.

## REPRODUCIBILITY STATEMENT

We provide detailed descriptions of our algorithms, mathematical formulations, training procedures, and evaluation settings. All datasets employed in this work are publicly available, including Pick-a-Pic, Rapidata, and VBench. Implementation details such as hyperparameters, batch sizes, learning rates, and model architectures are reported in the paper. In addition, extensive quantitative and qualitative results, ablation studies, and comparisons with baselines are included. These details ensure that the results presented in this paper can be reproduced by the research community.

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

SIPO: IMPORTANCE-SAMPLED DIRECT PREFERENCE OPTIMIZATION FOR STABLE DIFFUSION TRAINING

## A  NOTATION

| Symbol | Description | Equation |
|---|---|---|
| $x_0$ | Original data sample (e.g., image or video frame). | Eq. (1) |
| $x_t$ | Noisy sample at diffusion step $t$. | Eq. (1) |
| $\beta_t$ | Variance schedule parameter controlling noise strength. | Eq. (1) |
| $\epsilon \sim \mathcal{N}(0, I)$ | Gaussian noise injected during the forward process. | Eq. (3) |
| $\epsilon_\theta(x_t, t)$ | Predicted noise by the neural network at step $t$. | Eq. (3) |
| $\alpha_t, \bar{\alpha}_t$ | Noise accumulation coefficients. | Eq. (3) |
| $p_\theta(x_{t-1} \mid x_t)$ | Reverse diffusion distribution (model approximation). | Eq. (2) |
| $\mu_\theta, \Sigma_\theta$ | Predicted mean and variance of the reverse process. | Eq. (2) |
| $q(x_t \mid x_{t-1})$ | Forward noising distribution. | Eq. (1) |
| $w(t)$ | Importance weight at timestep $t$. | Eq. (5) |
| $\tilde{w}(t)$ | Clipped importance weight (with thresholding). | Eq. (11) |
| $w_\theta(x_0|c)$ | Importance weight ratio between model distribution and sampling distribution. | Eq. (A3) |
| $\beta$ | Temperature / regularization coefficient controlling the KL penalty. | Eq. (7), (16) |
| $L_{\mathrm{DM}}$ | Diffusion model training objective. | Eq. (3) |
| $L_{\mathrm{DPO}}$ | Direct Preference Optimization objective. | Eq. (9) |
| $L_{\mathrm{DPO\text{-}C\&M}}$ | DPO objective with Clipping & Masking. | Eq. (12) |
| $L_{\mathrm{SIPO}}$ | Importance-Sampled Direct Preference Optimization objective. | Eq. (16) |
| $p^*(x_0|c)$ | Target distribution after reward shaping. | Eq. (15), (A8) |
| $Z(c)$ | Normalization constant (partition function). | Eq. (8), (A7) |
| $D_{\mathrm{KL}}$ | Kullback–Leibler (KL) divergence. | Eq. (7) |
| $\sigma(\cdot)$ | Sigmoid function. | Eq. (6), (9), (12), (16) |

## B  MATHEMATICAL DERIVATIONS

We present detailed derivations of the key equations introduced in the main paper [see Eq. 18], to offer deeper insight into our method. To highlight the versatility of the proposed algorithm, we include both a flow-based implementation and its extension to large language models (LLMs).

### B.1  SIPO

The main optimization objective of RLHF is given by:

$$\max_{p_\theta} \mathbb{E}_{\mathbf{c} \sim \mathcal{D}_\mathbf{c}, \, \mathbf{x}_0 \sim p_\theta(\mathbf{x}_0|\mathbf{c})} [r(\mathbf{c}, \mathbf{x}_0)] - \beta \, \mathbb{D}_{\mathrm{KL}} [p_\theta(\mathbf{x}_0|\mathbf{c}) \,\|\, p_{\mathrm{ref}}(\mathbf{x}_0|\mathbf{c})]. \tag{A1}$$

We introduce *importance sampling*. According to its definition, the expectation under the target distribution $p_\theta(\mathbf{x}_0|\mathbf{c})$ can be rewritten using samples drawn from a behavior policy $q(\mathbf{x}_0|\mathbf{c})$:

$$\mathbb{E}_{\mathbf{x}_0 \sim p_\theta(\mathbf{x}_0|\mathbf{c})} [f(\mathbf{x}_0)] = \mathbb{E}_{\mathbf{x}_0 \sim q(\mathbf{x}_0|\mathbf{c})} \left[ \frac{p_\theta(\mathbf{x}_0|\mathbf{c})}{q(\mathbf{x}_0|\mathbf{c})} f(\mathbf{x}_0) \right]. \tag{A2}$$

To simplify notation, we define the importance weight $w_\theta(\mathbf{x}_0 \mid \mathbf{c})$ as:

$$w_\theta(\mathbf{x}_0 \mid \mathbf{c}) = \frac{p_\theta(\mathbf{x}_0 \mid \mathbf{c})}{q(\mathbf{x}_0 \mid \mathbf{c})}. \tag{A3}$$

Using this notation, the importance-sampled expectation becomes:

$$\mathbb{E}_{\mathbf{x}_0 \sim p_\theta(\mathbf{x}_0|\mathbf{c})} [f(\mathbf{x}_0)] = \mathbb{E}_{\mathbf{x}_0 \sim q(\mathbf{x}_0|\mathbf{c})} [w_\theta(\mathbf{x}_0 \mid \mathbf{c}) \, f(\mathbf{x}_0)]. \tag{A4}$$

We can rewrite the original objective as:

$$\max_{p_\theta} \quad \mathbb{E}_{\mathbf{c}\sim\mathcal{D}_\mathbf{c},\, \mathbf{x}_0\sim p_\theta(\mathbf{x}_0|\mathbf{c})} \left[ r(\mathbf{c}, \mathbf{x}_0) \right] - \beta\, \mathbb{D}_{\mathrm{KL}} \left[ p_\theta(\mathbf{x}_0|\mathbf{c}) \,\|\, p_{\mathrm{ref}}(\mathbf{x}_0|\mathbf{c}) \right]$$

$$= \max_{p_\theta} \quad \mathbb{E}_{\mathbf{c}\sim\mathcal{D}_\mathbf{c},\, \mathbf{x}_0\sim q(\mathbf{x}_0|\mathbf{c})} \left[ w_\theta \cdot r(\mathbf{c}, \mathbf{x}_0) \right] - \beta\, \mathbb{D}_{\mathrm{KL}} \left[ p_\theta(\mathbf{x}_0|\mathbf{c}) \,\|\, p_{\mathrm{ref}}(\mathbf{x}_0|\mathbf{c}) \right]$$

$$= \min_{p_\theta} \quad \mathbb{E}_{\mathbf{c}\sim\mathcal{D}_\mathbf{c},\, \mathbf{x}_0\sim q(\mathbf{x}_0|\mathbf{c})} \left[ \log\left( \frac{p_\theta(\mathbf{x}_0|\mathbf{c})}{p_{\mathrm{ref}}(\mathbf{x}_0|\mathbf{c}) \cdot \exp\left( \frac{w_\theta}{\beta} \cdot r(\mathbf{c}, \mathbf{x}_0) \right)} \right) \right] \quad \text{(A5)}$$

$$= \min_{p_\theta} \quad \mathbb{E}_{\mathbf{c}\sim\mathcal{D}_\mathbf{c},\, \mathbf{x}_0\sim q(\mathbf{x}_0|\mathbf{c})} \left[ \log\left( \frac{p_\theta(\mathbf{x}_0|\mathbf{c})}{\frac{1}{Z(\mathbf{c})} \cdot p_{\mathrm{ref}}(\mathbf{x}_0|\mathbf{c}) \cdot \exp\left( \frac{w_\theta}{\beta} \cdot r(\mathbf{c}, \mathbf{x}_0) \right)} \right) - \log Z(\mathbf{c}) \right]$$

Here, $w_\theta = \frac{p_\theta(x_0|c)}{q(x_0|c)}$ arises from importance sampling and corrects for the mismatch between the model distribution and the off-policy sampling distribution $q$.

The importance weight $w_\theta = \frac{p_\theta(x_0|c)}{q(x_0|c)}$ compensates for the discrepancy between the model distribution and the off-policy sampling distribution $q$, enabling effective training on pre-collected data. In parallel, the KL divergence term serves as a regularizer, preventing the learned model from deviating excessively from the reference model $p_{\mathrm{ref}}$. As noted in GRPO (Shao et al., 2024), this regularization is independent of the underlying data distribution, relying solely on the divergence between the current and reference models. Furthermore, TIS-DPO (Liu et al., 2024) demonstrates that this importance-sampling formulation provides an unbiased estimator for off-policy optimization.

As a result, we arrive at the following form of the objective:

$$\min_{p_\theta} \quad \mathbb{E}_{\mathbf{c}\sim\mathcal{D}_\mathbf{c},\, \mathbf{x}_0\sim q(\mathbf{x}_0|\mathbf{c})} \left[ \log\left( \frac{p_\theta(\mathbf{x}_0|\mathbf{c})}{\frac{1}{Z(\mathbf{c})} \cdot p_{\mathrm{ref}}(\mathbf{x}_0|\mathbf{c}) \cdot \exp\left( \frac{w_\theta}{\beta} \cdot r(\mathbf{c}, \mathbf{x}_0) \right)} \right) - \log Z(\mathbf{c}) \right], \quad \text{(A6)}$$

where

$$Z(\mathbf{c}) = \sum_{\mathbf{x}_0} p_{\mathrm{ref}}(\mathbf{x}_0 \mid \mathbf{c}) \cdot \exp\left( \frac{1+\epsilon}{\beta} \cdot r(\mathbf{c}, \mathbf{x}_0) \right). \quad \text{(A7)}$$

Here, $\epsilon$ is the clipping threshold used in the importance weight definition (see Eq. 12), and $Z(\mathbf{c})$ is a partition function.

After computing the importance weight $w_\theta$, a clipping operation is applied to constrain its value within the range $[1 - \epsilon,\, 1 + \epsilon]$. In reinforcement learning, extreme importance weights (either too large or too small) indicate significant distribution mismatch, which can harm training stability. This clipping strategy is a standard practice in major RLHF methods such as PPO and GRPO, and the same convention is followed here. In other words, the maximum value of $w_\theta$ is bounded by $1 + \epsilon$, where $\epsilon$ is typically a small fixed constant in practice. Therefore, using $1 + \epsilon$ as a surrogate for $w_\theta$ is both reasonable and consistent with empirical behavior.

The motivation for replacing $w_\theta$ with $1 + \epsilon$ is twofold: (1) it makes the partition function $Z(c)$ independent of $w_\theta$ and $p_\theta$, simplifying subsequent derivations; and (2) by assigning $Z(c)$ its maximal value, it ensures that $p^*$ in remains a valid probability in the range $[0, 1]$. This substitution is theoretically sound and supported by experimental results.

We further define the target distribution:

$$p^*(\mathbf{x}_0|\mathbf{c}) = \frac{1}{Z(\mathbf{c})} \cdot p_{\mathrm{ref}}(\mathbf{x}_0|\mathbf{c}) \cdot \exp\left( \frac{w_\theta}{\beta} \cdot r(\mathbf{c}, \mathbf{x}_0) \right), \quad \text{(A8)}$$

under which the optimization reduces to:

$$\min_{p_\theta} \quad \mathbb{E}_{\mathbf{c}\sim\mathcal{D}_\mathbf{c},\, \mathbf{x}_0\sim q(\mathbf{x}_0|\mathbf{c})} \left[ \log\left( \frac{p_\theta(\mathbf{x}_0|\mathbf{c})}{p^*(\mathbf{x}_0|\mathbf{c})} \right) - \log Z(\mathbf{c}) \right]$$

$$= \min_{p_\theta} \quad \mathbb{E}_{\mathbf{c}\sim\mathcal{D}_\mathbf{c}} \left[ \mathbb{D}_{\mathrm{KL}} \left( p_\theta(\mathbf{x}_0|\mathbf{c}) \,\|\, p^*(\mathbf{x}_0|\mathbf{c}) \right) - \log Z(\mathbf{c}) \right]. \quad \text{(A9)}$$

Since $Z(\mathbf{c})$ is independent of $p_\theta$, minimizing the KL divergence reduces to matching $p_\theta$ to $p^*$, allowing us to directly optimize toward the target distribution $p^*(\mathbf{x}_0 \mid \mathbf{c})$.

Based on the definition of $p^*(\mathbf{x}_0 \mid \mathbf{c})$, we can rearrange and express the reward as:

$$r(\mathbf{c}, \mathbf{x}_0) = \frac{\beta}{w_\theta} \cdot \left[ \log \left( \frac{p^*(\mathbf{x}_0|\mathbf{c})}{p_{\text{ref}}(\mathbf{x}_0|\mathbf{c})} \right) + \log Z(\mathbf{c}) \right]. \tag{A10}$$

We now derive the preference probability under the SIPO objective by substituting the reward expression into the Bradley–Terry model:

$$\mathcal{P}(\mathbf{x}_0^w \succ \mathbf{x}_0^l \mid \mathbf{c}) = \frac{\exp\left(r(\mathbf{c}, \mathbf{x}_0^w)\right)}{\exp\left(r(\mathbf{c}, \mathbf{x}_0^w)\right) + \exp\left(r(\mathbf{c}, \mathbf{x}_0^l)\right)}. \tag{A11}$$

Substituting Eq. equation A10 into the above, we obtain:

$$\mathcal{P}(\mathbf{x}_0^w \succ \mathbf{x}_0^l \mid \mathbf{c}) = \frac{\exp\left(\frac{\beta}{w_\theta^w}\left[\log\left(\frac{p^*(\mathbf{x}_0^w|\mathbf{c})}{p_{\text{ref}}(\mathbf{x}_0^w|\mathbf{c})}\right) + \log Z(\mathbf{c})\right]\right)}{\exp\left(\frac{\beta}{w_\theta^w}\left[\log\left(\frac{p^*(\mathbf{x}_0^w|\mathbf{c})}{p_{\text{ref}}(\mathbf{x}_0^w|\mathbf{c})}\right) + \log Z(\mathbf{c})\right]\right) + \exp\left(\frac{\beta}{w_\theta^l}\left[\log\left(\frac{p^*(\mathbf{x}_0^l|\mathbf{c})}{p_{\text{ref}}(\mathbf{x}_0^l|\mathbf{c})}\right) + \log Z(\mathbf{c})\right]\right)} \tag{A12}$$

By factoring out $\log Z(\mathbf{c})$ and simplifying, we get:

$$\mathcal{P}(\mathbf{x}_0^w \succ \mathbf{x}_0^l \mid \mathbf{c}) = \frac{1}{1 + \exp\left(\frac{\beta}{w_\theta^l} \log \frac{p^*(\mathbf{x}_0^l|\mathbf{c})}{p_{\text{ref}}(\mathbf{x}_0^l|\mathbf{c})} - \frac{\beta}{w_\theta^w} \log \frac{p^*(\mathbf{x}_0^w|\mathbf{c})}{p_{\text{ref}}(\mathbf{x}_0^w|\mathbf{c})}\right)} \tag{A13}$$

$$= \sigma\left(\frac{\beta}{w_\theta^w} \log \frac{p^*(\mathbf{x}_0^w \mid \mathbf{c})}{p_{\text{ref}}(\mathbf{x}_0^w \mid \mathbf{c})} - \frac{\beta}{w_\theta^l} \log \frac{p^*(\mathbf{x}_0^l \mid \mathbf{c})}{p_{\text{ref}}(\mathbf{x}_0^l \mid \mathbf{c})}\right). \tag{A14}$$

Thus, we arrive at the SIPO loss, which mirrors the form of the DPO loss shown above, but incorporates off-policy correction via importance weights:

$$\mathcal{L}_{\text{S-DPO}}(\theta) = -\mathbb{E}_{(\mathbf{c}, \mathbf{x}_0^w, \mathbf{x}_0^l) \sim \mathcal{D}} \left[ \log \sigma \left( \frac{\beta}{w_\theta^w} \log \frac{p_\theta(\mathbf{x}_0^w \mid \mathbf{c})}{p_{\text{ref}}(\mathbf{x}_0^w \mid \mathbf{c})} - \frac{\beta}{w_\theta^l} \log \frac{p_\theta(\mathbf{x}_0^l \mid \mathbf{c})}{p_{\text{ref}}(\mathbf{x}_0^l \mid \mathbf{c})} \right) \right]. \tag{A15}$$

To improve stability, we define a shared clipped inverse importance weight:

$$\tilde{w}_\theta = \max\left(\frac{1}{w_\theta^w}, \frac{1}{w_\theta^l}\right), \quad \tilde{w}_\theta \leftarrow \text{clip}(\tilde{w}_\theta, 1 - \epsilon, 1 + \epsilon). \tag{A16}$$

We then use $\tilde{w}_\theta$ to scale the entire preference score difference, leading to the final SIPO loss:

$$\mathcal{L}_{\text{SIPO}}(\theta) = -\mathbb{E}_{(\mathbf{c}, \mathbf{x}_0^w, \mathbf{x}_0^l) \sim \mathcal{D}} \left[ \log \sigma \left( \frac{\beta}{\tilde{w}_\theta} \cdot \left[ \log \frac{p_\theta(\mathbf{x}_0^w \mid \mathbf{c})}{p_{\text{ref}}(\mathbf{x}_0^w \mid \mathbf{c})} - \log \frac{p_\theta(\mathbf{x}_0^l \mid \mathbf{c})}{p_{\text{ref}}(\mathbf{x}_0^l \mid \mathbf{c})} \right] \right) \right]. \tag{A17}$$

### B.2 Advantages of the Modified Objective

Compared with the original DPO objective (which only uses the term $\log \frac{p}{p_{\text{ref}}}$), the modified objective

$$A(p) = \frac{q}{p} \log \frac{p}{p_{\text{ref}}}, \quad (p = p_\theta(x|c))$$

has several notable advantages:

1. **Adaptive step size with priority on "low-confidence" samples.** The derivative is

$$\frac{\partial A}{\partial p} = q \cdot \frac{1 - \log(p/p_{\text{ref}})}{p^2}.$$

When $p \ll p_{\text{ref}}$, the gradient is large ($\propto 1/p^2$), which enforces stronger updates. When $p \gg p_{\text{ref}}$, the gradient can even become negative, meaning updates will self-regulate and prevent runaway increases in probability. In contrast, the original DPO gradient decays only as $1/p$, lacking this adaptivity.

2. **Soft upper bound / two-sided constraint on probabilities.** As $p$ grows beyond $p^* = e\,p_{\text{ref}}$, the objective reaches its maximum and then decreases. This effectively regularizes the optimization, encouraging the model to keep probabilities near the reference instead of endlessly increasing them, mitigating issues like "reward hacking."

3. **Improved robustness against bias.** The scaling factor $q/p$ implicitly downweights samples where the model is already highly confident (large $p$), while upweighting samples where $p$ is small. This balances the influence of samples and stabilizes training, especially when good samples come from diverse sources.

4. **Better numerical stability.** The growth rate of $\log(\cdot)$ ensures bounded changes. In extreme cases ($p \to \infty$), the effective update magnitude is well controlled, which improves overall training stability compared to the original DPO.

## B.3 APPLICABILITY OF SIPO TO LARGE LANGUAGE MODELS

Although our experiments are conducted in the diffusion model setting, it is evident that SIPO is modality-agnostic and can be readily applied to large language models (LLMs); we include such results in a later section to demonstrate its effectiveness.

## B.4 DIFFUSION-BASED SIPO OBJECTIVE

To adapt the SIPO objective to the diffusion setting, we follow the formulation introduced in Diffusion-DPO. We express the preference score difference in terms of the diffusion model's transition probabilities. This leads to the following loss:

$$\mathcal{L}_{\text{SIPO}}(\theta) \leq -\mathbb{E}_{(\mathbf{x}_0^w, \mathbf{x}_0^l) \sim \mathcal{D}, \, t \sim \mathcal{U}(0,T)} \left[ \log \sigma \left( \frac{\beta T}{\tilde{w}_\theta} \cdot \left[ \log \frac{p_\theta(\mathbf{x}_{t-1}^w | \mathbf{x}_t^w)}{p_{\text{ref}}(\mathbf{x}_{t-1}^w | \mathbf{x}_t^w)} - \log \frac{p_\theta(\mathbf{x}_{t-1}^l | \mathbf{x}_t^l)}{p_{\text{ref}}(\mathbf{x}_{t-1}^l | \mathbf{x}_t^l)} \right] \right) \right] \tag{A18}$$

**Post-Training Alignment of Language Models** Large language models (LLMs) are typically aligned with human preferences through Reinforcement Learning from Human Feedback (RLHF) Ouyang et al. (2022), which trains a reward model from human comparisons followed by fine-tuning with reinforcement learning. Direct Preference Optimization (DPO) Rafailov et al. (2023) has emerged as a simpler alternative by reframing alignment as supervised preference learning, removing the need for sampling or reward modeling during training. Building on DPO, recent variants improve reasoning, scalability, and feedback efficiency: StepDPO Lai et al. (2024) introduces step-level supervision (unsuitable for diffusion models); VPO and SimPO Cen et al. (2024) enable online updates via implicit rewards; and KTO Ethayarajh et al. (2024), inspired by prospect theory Kahneman & Tversky (1992), optimizes utility with binary feedback. Other approaches include iterative optimization over reasoning steps Xie et al. (2024); Pang et al. (2024), dual-LoRA-based stabilization in Online DPO Qi et al. (2024), self-play in SPO Swamy et al. (2024), likelihood calibration in SLiC Zhao et al. (2022), and ranking-based alignment in RRHF Yuan et al. (2023).

**Preference Optimization for Diffusion Models** Recent work extends human preference alignment to diffusion models, drawing inspiration from RLHF in language models. Early models like Stable Diffusion used curated data without human feedback. Diffusion DPO Wallace et al. (2024) adapts DPO using the ELBO as a proxy, improving SDXL with large-scale preference data. RL-based methods such as DPOK Fan et al. (2023) and DDPO Black et al. (2023) apply KL-regularized policy gradients or actor-critic training, often requiring constrained generation for stability. IPO Yang et al. (2025) reduces policy-data mismatch via iterative reward updates, though the process is complex. D3PO Yang et al. (2024a) avoids reward modeling by directly optimizing on pairwise feedback.

Gradient-based methods like VADER Prabhudesai et al. (2024) and DRaFT Clark et al. (2024) align outputs via backpropagation through reward models. SPIN-Diffusion Yuan et al. (2024) removes human labels entirely, using self-play with automated preferences to achieve strong results. Together, RL-based Fan et al. (2023); Black et al. (2023), direct optimization Wallace et al. (2024); Yang et al. (2024a), iterative Yang et al. (2025), gradient-based Prabhudesai et al. (2024); Clark et al. (2024), and self-play Yuan et al. (2024) methods form a growing toolkit for aligning diffusion models with human preferences.

## C  RELATED WORK

**Diffusion models** Diffusion-based generative models have become a leading approach for high-quality image and video synthesis. Early methods built on score-based Langevin dynamics (Song & Ermon, 2019), later formalized as DDPMs (Ho et al., 2020a) and unified via SDEs for improved likelihood estimation and generation (et al., 2021b). Sampling speed was improved through DDIM (Jiaming Song, 2021) and distillation (Salimans & Ho, 2022), while realism benefited from classifier guidance, architectural advances (Dhariwal & Nichol, 2021), and classifier-free methods (et al., 2021a). Latent Diffusion Models (et al., 2022b) enhanced efficiency, and further refinements improved FID (Karras et al., 2022). In video, diffusion models have been extended to capture temporal dynamics (et al., 2022a), achieving strong results in generation, prediction, and interpolation (et al., 2022c; Ho et al., 2022; Singer et al., 2022). Recent text-to-video methods leverage transformer-based architectures (Brooks et al., 2024; Yang et al., 2024b; Chen et al., 2024a), often adapting text-to-image backbones with temporal attention (Blattmann et al., 2023; Wang et al., 2023; Guo et al., 2023). Lightweight modules (Guo et al., 2023) enable plug-and-play personalization. Recent works report state-of-the-art quality and scalability (Zhang et al., 2023; Chen et al., 2024a), with DiT-based models pushing further (Brooks et al., 2024; Yang et al., 2024b). Despite advances in fidelity, consistency, and diversity (Chen et al., 2023; Esser et al., 2023; et al., 2022a; Liu et al., 2023a; Ruan et al., 2023), aligning outputs with nuanced human preferences remains a key challenge (Ho et al., 2022; Wu et al., 2021).

**Preference Optimization for Diffusion Models** Recent work extends human preference alignment to diffusion models, drawing inspiration from RLHF in language models. Early models like Stable Diffusion used curated data without human feedback. Diffusion DPO (Wallace et al., 2024) adapts DPO using the ELBO as a proxy, improving SDXL with large-scale preference data. RL-based methods such as DPOK (Fan et al., 2023) and DDPO (Black et al., 2023) apply KL-regularized policy gradients or actor-critic training, often requiring constrained generation for stability. IPO (Yang et al., 2025) reduces policy-data mismatch via iterative reward updates, though the process is complex. D3PO (Yang et al., 2024a) avoids reward modeling by directly optimizing on pairwise feedback. Gradient-based methods like VADER (Prabhudesai et al., 2024) and DRaFT (Clark et al., 2024) align outputs via backpropagation through reward models. SPIN-Diffusion (Yuan et al., 2024) removes human labels entirely, using self-play with automated preferences to achieve strong results. Together, RL-based (Fan et al., 2023; Black et al., 2023), direct optimization (Wallace et al., 2024; Yang et al., 2024a), iterative (Yang et al., 2025), gradient-based (Prabhudesai et al., 2024; Clark et al., 2024), and self-play (Yuan et al., 2024) methods form a growing toolkit for aligning diffusion models with human preferences.

## D  EXPERIMENTAL SETUP DETAILS

For completeness, we summarize the experimental configurations used in our work.

**Models and Dataset.** We conduct experiments on both text-to-video and text-to-image generation models, comparing standard Diffusion-DPO with our improved method, SIPO. For text-to-video, we base our experiments on three backbone models: CogVideoX-2B, CogVideoX-5B (Yang et al., 2024b), and Wan-1.3B (Wan et al., 2025). For each prompt, we generate multiple candidate videos and collect human annotations. After filtering, we construct a dataset of 10,000 high-quality preference pairs to train our methods. For text-to-image, we conduct experiments on Stable Diffusion 1.5 (SD1.5) and Stable Diffusion XL-1.0 (SDXL). We use the Pick-a-Pic dataset (Kirstain et al., 2023), which consists of pairwise human preferences for images generated by SDXL-beta and Dreamlike,

| Name | Description | Value |
|------|-------------|-------|
| lr | learning rate | $2 \times 10^{-5}$ |
| optimizer | optimizer type | Adam |
| bs | batch size per GPU | 4 (video), 1 (image) |
| $G$ | gradient accumulation steps | 8 (video), 64 (image) |
| $\beta$ | temperature | 2 (DPO, video), 0.02 (SIPO/DPO-C&M, video); 5 (DPO, image), 1 (SIPO/DPO-C&M, image) |
| GPUs | number of GPUs | $16 \times$ A100 |
| precision | mixed precision | fp16 |

Table A1: Training hyperparameters for text-to-video and text-to-image experiments.

a fine-tuned version of SD1.5. This dataset provides a diverse and challenging benchmark for evaluating preference-based alignment methods in text-to-image generation.

**Training Details.** In the text-to-video setting, training is performed on 16 NVIDIA A100 GPUs, using a batch size of 4 with gradient accumulation of 8 and a fixed learning rate of $2 \times 10^{-5}$. The temperature $\beta$ is assigned a value of 2 for Diffusion-DPO, while both SIPO and DPO-C&M use 0.02. In the text-to-image case, we likewise employ 16 A100 GPUs, but with a batch size of 1 and gradient accumulation of 64. Here, $\beta$ is set to 5 for Diffusion-DPO, and to 1 for SIPO and DPO-C&M.

**Evaluation.** For text-to-video generation, we evaluate post-training performance primarily using VBench Huang et al. (2024), a large-scale benchmark specifically designed for video generation. VBench disentangles the evaluation into 16 dimensions that span aspects such as visual quality, motion consistency, and semantic alignment. Following standard practice, we report three aggregated metrics: Total Score, Quality Score, and Semantic Score.

For text-to-image generation, we rely on the Pick-a-Pic test split as well as the HPSv2 benchmark. Evaluation is carried out using several human-alignment reward models: PickScore Kirstain et al. (2023), ImageReward Xu et al. (2023), and HPSv2 Wu et al. (2023). Each of these models is trained on large-scale human-annotated preference datasets. In addition, we consider Aesthetic scores (Schuhmann, 2022) to capture perceptual quality. We report the average score across all metrics to reflect overall fidelity and human preference alignment.

# E  HUMAN EVALUATION PROTOCOL

In order to evaluate the perceived quality and preference alignment of generated results, we perform a controlled human study based on pairwise comparisons between models. This section describes the prompt design, annotator recruitment, annotation procedure, quality control, and the computation of the final metrics.

## E.1  PROMPT DESIGN

We construct a set of **200 text prompts** that covers a broad and diverse range of scenarios. The goal is to probe both semantic fidelity and visual quality under different types of compositional complexity. In particular, the prompt set explicitly includes the following categories:

- **Humans and characters**, such as prompts that specify appearance, pose, simple activities, or interactions.

- **Animals**, including both single and multiple animals with clearly specified attributes.

- **Objects and attributes**, where color, size, style, and spatial relations are described in the text.

- **Single object versus multi object scenes**, which differ in the number of entities and the complexity of the relationships.

- **Food and daily items**, which test recognizable everyday objects and presentation quality.

- **Vehicles and transportation scenes**, such as cars, trains, airplanes, and related contexts.

The prompts are written in a templated yet varied manner in order to balance coverage and control. For each prompt, two models under comparison generate outputs using the same sampling configuration, for example the same number of inference steps and the same guidance scale. No manual cherry picking or qualitative filtering of the generated outputs is performed at this stage. For each prompt, the two outputs are then bundled into a single pair that will be presented to annotators.

### E.2 ANNOTATOR RECRUITMENT AND BACKGROUND

We recruit **12 annotators** for the study. Their basic demographic information is as follows:

- **Gender**: 6 male and 6 female.
- **Age**: between **21** and **25** years old.
- **Education**: all annotators hold, or are in the process of obtaining, a **bachelor's degree**.

None of the annotators participate in the development or training of the models, and they do not have access to implementation details that would reveal which model corresponds to which research method. During the study, model identities are represented only by anonymized identifiers such as "Model A" and "Model B". This design is intended to keep the evaluation blind with respect to the underlying method and to reduce potential bias.

Before the main annotation phase, we provide a short written guideline that explains the goal of the study and illustrates a few example comparisons. Annotators are encouraged to ask clarification questions about the instructions. No examples from the actual evaluation set are used during this guideline phase.

### E.3 ANNOTATION PROCEDURE

To obtain multiple independent judgments for each comparison, we split the 12 annotators into **3 groups** of 4 people. Each group independently annotates the full set of 200 prompt and pair samples, which means that each comparison is evaluated three times, once per group.

For each prompt, annotators are shown the two generated results side by side, in an interface that randomizes the left and right positions for every sample. The textual prompt is displayed above the two outputs. At no point is the identity of the underlying model revealed.

Annotators receive the following instruction:

> Given the text prompt, please choose the result that is overall better. You should consider both (1) how well the result matches the text description and (2) its overall visual quality, including clarity and absence of artifacts. If you consider the two results essentially indistinguishable with respect to these criteria, please choose "tie".

Each annotator works individually and makes decisions without discussion with other annotators. Within each group, we first aggregate the 4 individual labels by **majority vote** to obtain a single group level decision for each prompt pair. In case of a tie among the 4 annotators, the group level label is recorded as a tie. After this step, we have three group level labels for each comparison.

To obtain the final label for each prompt pair, we again apply majority vote across the 3 groups. If all three group level labels agree, that label is used directly. If there is partial disagreement, but one label still appears at least twice, that label is taken as the final outcome. If the three group level labels do not produce a strict majority, the comparison is counted as a final tie.

This two stage aggregation, first within groups and then across groups, is used to reduce the influence of outlier judgments and to check the consistency of preferences across different subsets of annotators.

### E.4 QUALITY CONTROL AND FAIRNESS

We apply several basic quality control measures to ensure that the collected labels are reliable:

- We discard responses where an annotator clearly does not follow the instructions. Typical patterns include missing answers on many items or selecting the same side for almost all comparisons regardless of content. When such issues are detected early, the corresponding pairs are reassigned within the group.

- The identities of the models are fully anonymized throughout the study, and the position of each model in the side by side view is independently randomized for every sample. This design reduces potential positional bias and prevents annotators from associating a model with a fixed position.

- All prompts are designed to be neutral and to avoid sensitive or offensive content. This helps keep the task focused on quality and alignment rather than on moral or cultural judgments.

- The order of the 200 comparisons is randomized separately for each group. Annotators are allowed to take breaks as needed so that fatigue does not systematically affect the later part of the evaluation.

In addition to aggregation by majority vote, we also monitor basic agreement statistics across annotators and across groups in order to confirm that the task is well posed and that the resulting labels are not dominated by random guessing.

### E.5 AGGREGATION AND REPORTED METRICS

For each pair of models, we compute a win, tie, and loss count over the 200 prompts based on the final aggregated labels described above. In the main paper, we report **win rates** as the primary human evaluation metric. A win contributes one count to the corresponding model, a loss contributes zero, and a tie contributes one half to each model. The final win rate is obtained by dividing the total number of effective wins by the total number of comparisons.

This evaluation protocol combines a diverse and balanced prompt set, a blinded and randomized A/B presentation, 12 annotators split into 3 independent groups, and a two stage majority vote aggregation scheme. Together, these design choices aim to make the human evaluation procedure fair, robust, and reproducible, and to provide a reliable complement to the automatic metrics reported in the main text.

## F  MORE RESULTS

### F.1  SENSITIVITY TO PRUNING THRESHOLD AND CLIPPING PARAMETER $\epsilon$

In this section, we analyze the sensitivity of our method to the pruning threshold and the clipping parameter $\varepsilon$. Our choice of the pruning threshold is mainly motivated by two considerations. First, as visualized in Fig. 1 (corresponding to Fig. 8 in the main paper), when the importance weight threshold is set to 0.9, roughly the first $\sim 60$ timesteps are pruned. Our experiments show that discarding these low-signal, high-variance timesteps consistently stabilizes DPO training. Second, this choice is in line with common practice in PPO/GRPO, where clipping ratios of the form $[1 - \varepsilon, 1 + \varepsilon]$ with $\varepsilon \in \{0.1, 0.2\}$ are widely used as heuristic defaults; we similarly treat the pruning threshold as a robust heuristic rather than a finely tuned hyperparameter.

To directly study the sensitivity of the clipping parameter $\varepsilon$, we additionally train SD1.5 on the Pick-a-Pic v1 dataset (evaluated on the Pick-a-Pic test set) and SDXL on the Pick-a-Pic v2 dataset (evaluated on the HPSv2 test set to avoid contamination of the Pick-a-Pic test set), using HPSv2 score as the evaluation metric. We sweep $\varepsilon \in \{0.05, 0.10, 0.20, 0.30, 0.40, 0.50\}$. The results are summarized in Table A2.

From Table A2, we observe that $\varepsilon = 0.10$ achieves the highest score for both SDXL and SD1.5, with $\varepsilon = 0.20$ being very close. When $\varepsilon = 0.05$, too many timesteps are effectively clipped, leading to under-training and worse performance. As $\varepsilon$ increases beyond 0.20, relaxing the constraint allows larger importance ratios, which increases the variance of the gradient estimates and causes a gradual performance drop, consistent with standard importance sampling theory. Importantly, SD1.5 and SDXL exhibit the same qualitative trend across $\varepsilon$, suggesting that SIPO is not overly sensitive to the specific dataset or backbone.

Table A2: Sensitivity of SIPO to the clipping parameter $\varepsilon$ on SDXL and SD1.5, measured by HPSv2 score. "Baseline" denotes the pretrained model without preference alignment.

| Model | Baseline | 0.05 | 0.10 | 0.20 | 0.30 | 0.40 | 0.50 |
|---|---|---|---|---|---|---|---|
| SDXL | 0.2812 | 0.2853 | 0.2889 | 0.2885 | 0.2842 | 0.2833 | 0.2825 |
| SD1.5 | 0.2699 | 0.2720 | 0.2763 | 0.2759 | 0.2735 | 0.2718 | 0.2691 |

### F.2 ONLINE LEARNING.

We evaluate suitability for online and iterative training by adopting the IPO protocol (Yang et al., 2025), in which each round samples 3,000 prompts, ranks paired outputs using a fixed reward model (Liu et al., 2025), and trains for 20 epochs before proceeding to the next batch. This procedure is repeated for 10 iterations without human intervention. As shown in Fig. A1, Diffusion-DPO exhibits clear performance degradation over rounds, attributed to reward hacking and distributional drift. In contrast, SIPO maintains stable or slightly improving performance, supported by implicit update control that mitigates collapse when reward signals deteriorate. On average, SIPO out-

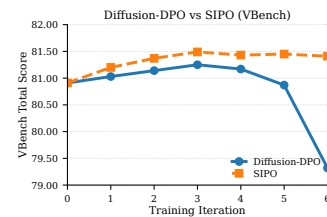

Figure A1: Comparison of Diffusion-DPO and SIPO on vbench over training iterations.

performs Diffusion-DPO by over 15% in win rate after the final round. Moreover, SIPO shows significantly lower variance across runs, indicating better training stability under non-stationary conditions. These results underscore SIPO's robustness and effectiveness in iterative preference optimization settings.

## G QUALITATIVE COMPARISON OF VISUAL OUTPUTS

To further evaluate the effectiveness of SIPO, we present qualitative comparisons of generated video outputs across multiple aspects, including text alignment, motion continuity, and style fidelity.

As illustrated in Fig. A2 and Fig. A3, SIPO significantly improves upon the base model (`Wan-1.3B`) in overall aesthetic quality. For instance, when generating stylized prompts such as "a cartoon dog" or "a cyberpunk panda," SIPO produces videos that better adhere to the intended artistic style compared to the base model.In terms of motion consistency, SIPO enhances the temporal dynamics of the generated videos. While outputs from the base model tend to be static or lack coherent motion, SIPO produces more fluid and plausible motion sequences.We also compare SIPO with Diffusion-DPO in Fig. A4. Across different evaluation axes—semantics, motion, and style—our method demonstrates consistent visual advantages, highlighting the effectiveness of our modality-agnostic preference optimization approach.

While SIPO significantly improves the stability and alignment quality of diffusion-based preference optimization, several limitations remain. The method relies on proxy importance weights derived from approximated reverse transitions, which may not fully capture dynamic changes in model behavior during training, limiting its responsiveness in highly non-stationary or long-horizon scenarios. Although SIPO effectively corrects off-policy bias, it assumes access to high-quality offline preference data and may degrade in settings with noisy or sparse feedback. While we demonstrate strong generalization to both score-based and flow-based diffusion models, extending SIPO to complex multimodal tasks (e.g., video-audio-text generation) remains an open challenge. Additionally, although our preliminary experiments show that SIPO is applicable to large language models (LLMs) and particularly suitable for online or continual learning, its effectiveness in such dynamic settings has not been fully explored. Investigating SIPO under streaming feedback, evolving preference distributions, or reinforcement-style update regimes is a promising direction. Finally, the extra computational overhead from importance weighting and masking, while moderate, may impact scalability to very large datasets or ultra-high-resolution generation tasks. Future work may explore adaptive weighting strategies, online preference data integration, and hybrid objectives combining SIPO with supervised fine-tuning or reinforcement learning techniques.

## H  USE OF LLMS

In this work, we made limited use of large language models (LLMs) to assist with peripheral tasks that do not affect the core methodology or experimental outcomes. Specifically, LLMs were employed to facilitate the construction of text prompts used for generating synthetic training data. These prompts served only as auxiliary input formats and did not introduce additional information or biases beyond what was already designed in our experimental setup.

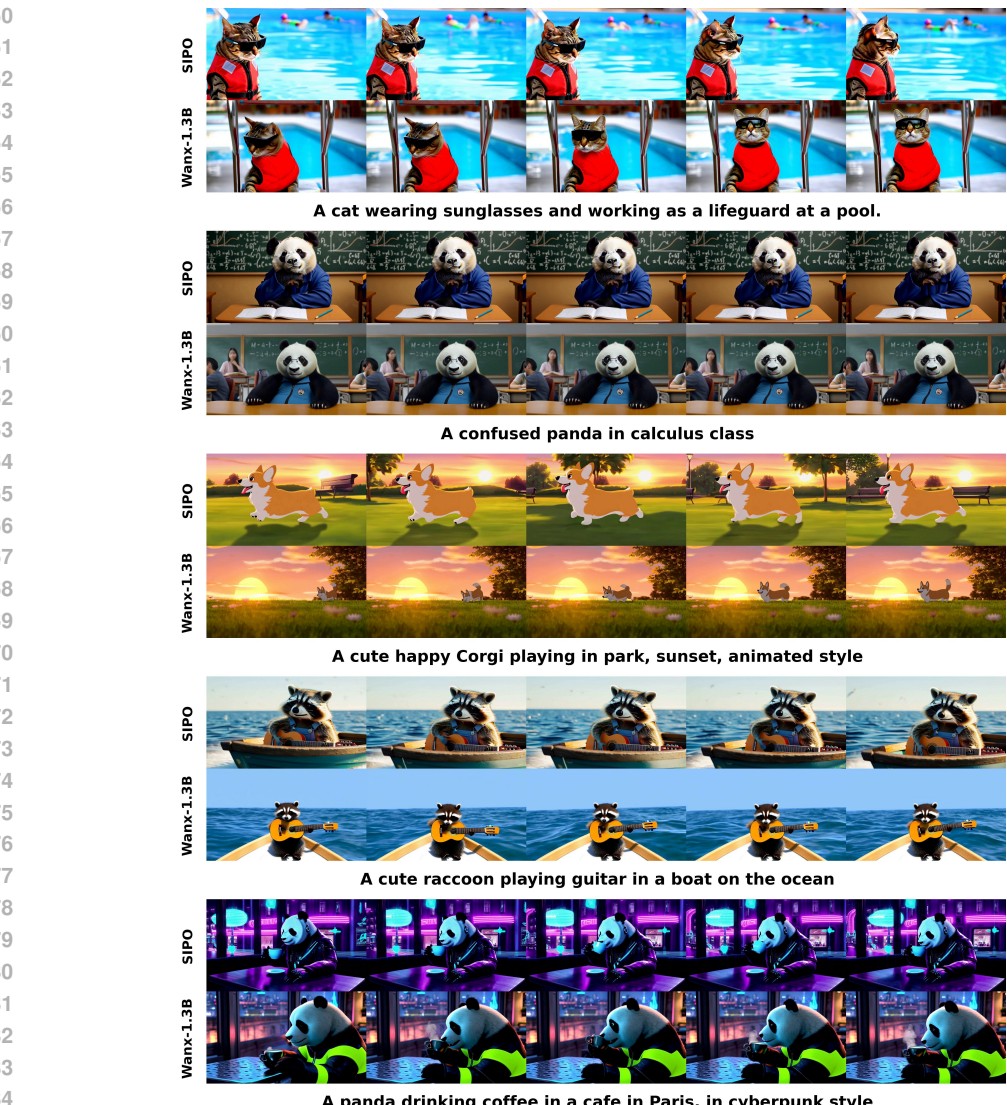

Figure A2: Side-by-side visual comparison of the SIPO-optimized model and the Wanx-1.3B base model.

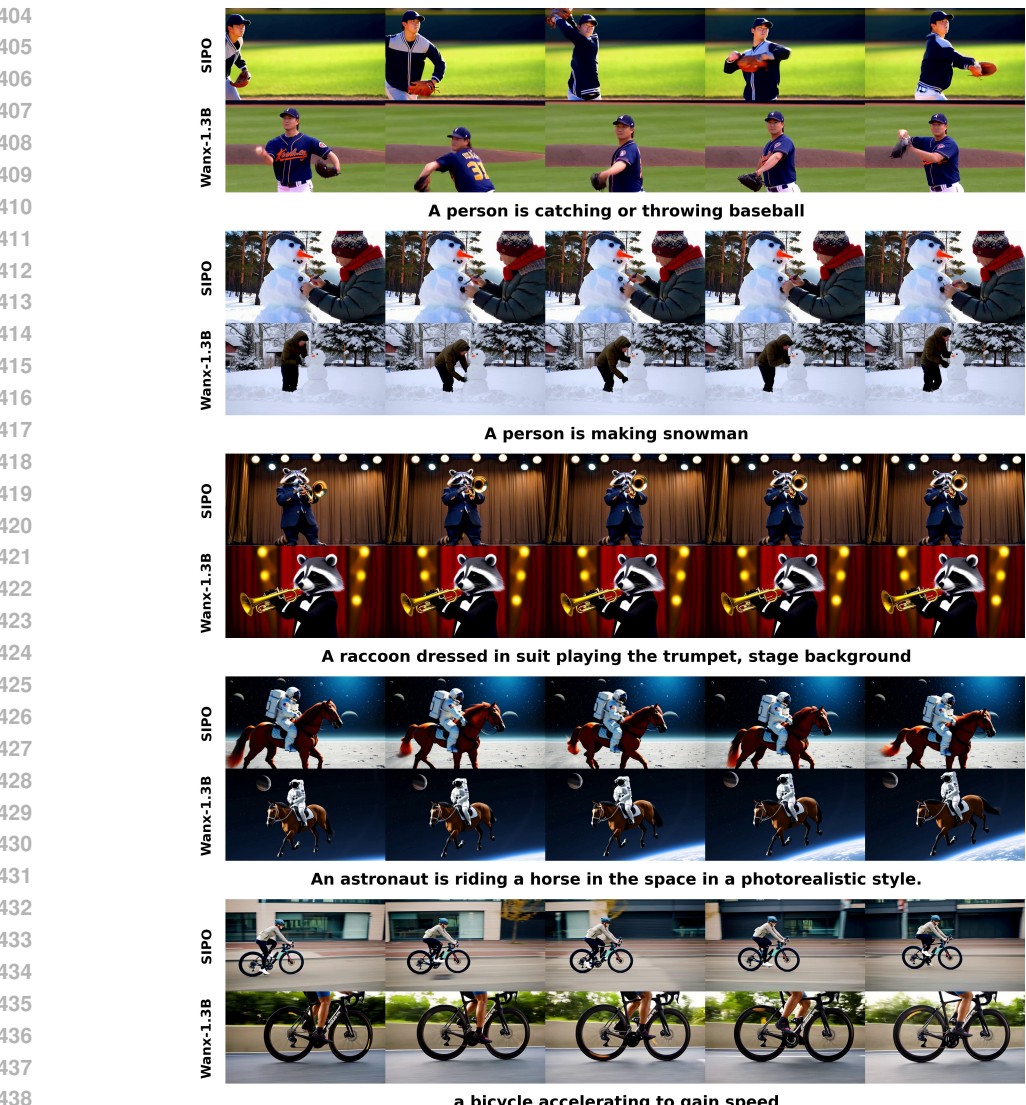

Figure A3: Side-by-side visual comparison of the SIPO-optimized model and the Wanx-1.3B base model.

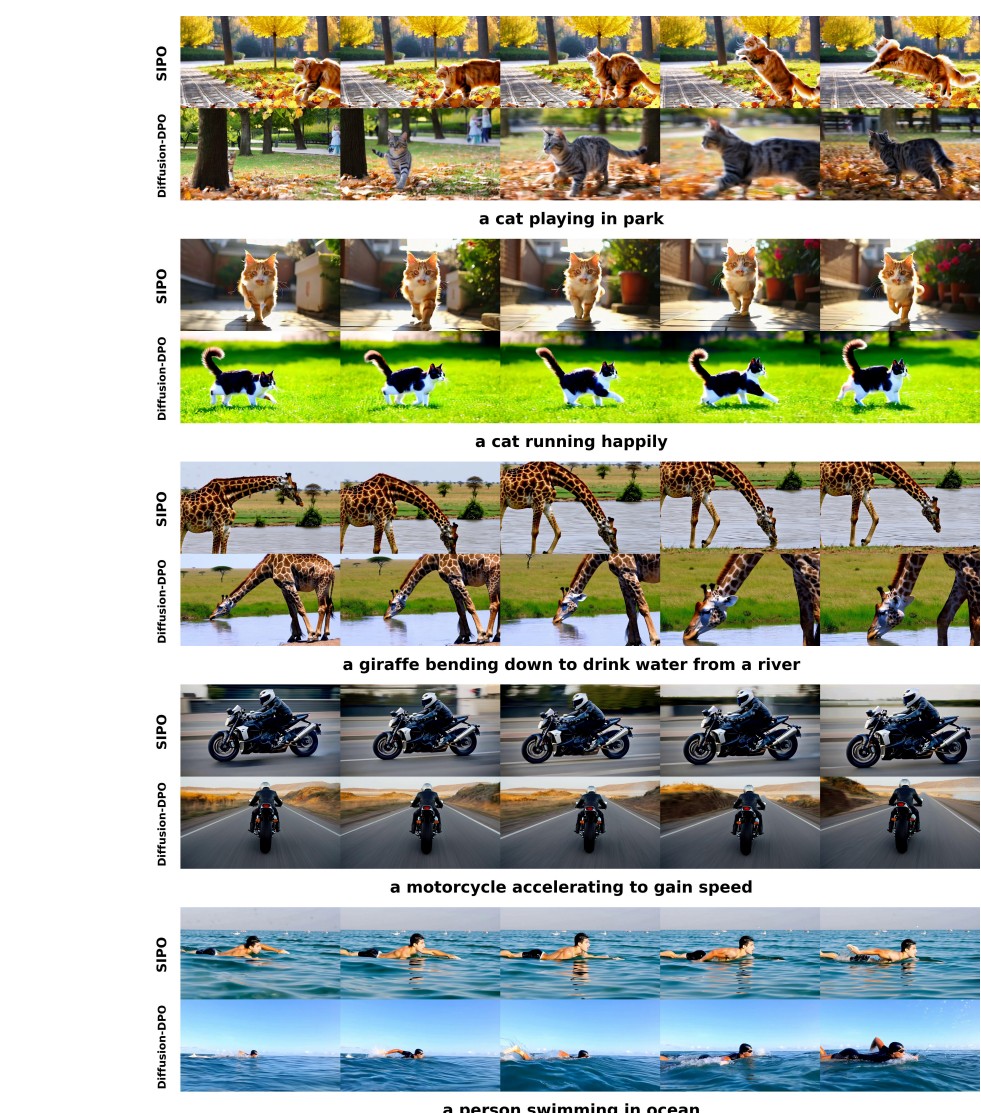

Figure A4: Side-by-side visual comparison of outputs from the Wanx-1.3B model after optimization with SIPO (up) and Diffusion-DPO (down).

