# OpenReview forum: "SIPO: Stabilized and Improved Preference Optimization for Aligning Diffusion Models"
_ICLR.cc/2026/Conference — Submitted to ICLR 2026_

### Official Review · Reviewer_X295 · 2025-10-16

**Soundness:** 2
**Presentation:** 3
**Contribution:** 2
**Rating:** 4
**Confidence:** 3

**Summary:**

The paper proposes Stabilized and Improved Preference Optimization (SIPO), a framework designed to address two fundamental challenges in applying Direct Preference Optimization (DPO) to diffusion models: training instability and off-policy bias. The authors first conduct a systematic analysis of the diffusion process, identifying that training instability is primarily caused by high gradient variances originating from early time steps that have low importance weights.

**Strengths:**

**Quality**

1. Stability and Robustness.
The method demonstrates high training quality by providing a smoother, more stable training loss curve (Figure 1b) and consistently improving test accuracy without the late-stage degradation (reward hacking) seen in Diffusion-DPO (Figure 1c, 8a, 8b). Crucially, SIPO shows remarkable robustness to the $\beta$ hyperparameter compared to the high sensitivity of Diffusion-DPO (Figure 1a)

2. Extensive Benchmarking.
The experiments are high-quality and comprehensive, validating SIPO across a diverse set of large-scale models and tasks, including both popular image generators (SDXL) and challenging video generation models (CogVideoX, Wan2.1-1.3B, FLUX-dev).


**Clarity**

1. Clear Problem Framing. The paper is clear in framing the work, articulating the two core challenges (instability and off-policy bias) and immediately tying them to the need for timestep-aware alignment.

2. Logical Flow. The introduction of the method is logically motivated by the preceding analysis on the reward dynamics and importance weights across different time steps (Figure 2 and 4), providing a foundation for the technical solution.

**Weaknesses:**

1. Missing Analysis on Late-Stage Instability.

The paper states that "early and late stages introduce instability" and that "preference signals are more informative in middle-to-late timesteps". However, the core analysis focuses heavily on early timesteps being problematic due to low importance weights (Figure 4, low weight up to $t \approx 63$). The paper is missing an explicit, corresponding analysis of why the very late timesteps, e.g., $t>900$ might also be unstable or uninformative, and how SIPO's mechanisms specifically address this tail-end instability.


2. Baselines. The baselines are not sufficient. More SOTA baselines are encouraged to compare with, such as SPPO [1] and RainbowPA [2].

[1] Bridging SFT and DPO for Diffusion Model Alignment with Self-Sampling Preference Optimization. arXiv:2410.05255, 2025.

[2] Diffusion-RainbowPA: Improvements Integrated Preference Alignment for Diffusion-based Text-to-Image Generation. Transactions on Machine Learning Research, 2025.

**Questions:**

1. Baselines. See W2.

2. Detailed Video Metrics.

The work explicitly addresses video generation challenges like maintaining temporal coherence. How about a breakdown of VBench results using metrics that specifically quantify temporal quality (e.g., motion smoothness, temporal consistency) in the main results section. This would provide stronger evidence for the successful alignment of video models compared to the general video quality metrics currently presented.

---

> ### Author Response · Authors · 2025-11-21
> **Response to Reviewer X295**
>
> >**Weaknesses 1: Missing Analysis on Late-Stage Instability.**
>
> We thank the reviewer for the very helpful comment. Our actual claim is that **middle-to-late timesteps are more informative than very early timesteps**, and this is also how we phrase it in the introduction. The statement that “early and late stages introduce instability” at the end of the introduction is a writing mistake. What we intended to express is that very late timesteps have very small noise and therefore provide only limited additional gain for DPO training, rather than being intrinsically unstable. Because the impact of very late timesteps is relatively minor, we revised our conclusion before submission to no longer discuss them separately, but we failed to fully update this sentence accordingly.
>
> To clarify this, we additionally run an ablation where we train SD1.5 DPO under three timestep ranges and evaluate on HPSv2, Pick-a-Pic score, ImageReward, and Aesthetic, and we also report the SD1.5 pretrained baseline:
>
> | Method / Timesteps used for training              | HPSv2 ↑ | Pick-a-Pic Score ↑ | ImageReward ↑ | Aesthetic ↑ |
> |-------------------------------------------|--------:|--------------------:|--------------:|------------:|
> | SD1.5 Pretrained                                               |  0.2699   |       21.1893         |     0.8159      |   5.7691    |
> | DPO, all timesteps [0, T]                                    |  0.2753   |        21.8866        |      1.0495      |     5.8918     |
> | DPO, remove early timesteps (t < 60)                |  **0.2771**   |        **21.8892**         |     1.1135       |   6.0047     |
> | DPO, remove early + late (t < 60, t > 900)         | 0.2769   |       21.8891         |    **1.1138**      |   **6.0052**      |
>
> From the table, we observe that removing the late timesteps (t > 900) on top of removing the early timesteps changes the metrics only marginally, while removing the early timesteps brings the main improvement over the pretrained baseline. This supports that the early timesteps are problematic, whereas the very late timesteps have relatively minor impact. We will correct the misleading sentence in the introduction and include this ablation in the revised manuscript. We again thank the reviewer for pointing out this issue.
>
> >**Weaknesses 2: Missing Analysis on Late-Stage Instability.Baselines. The baselines are not sufficient. More SOTA baselines are encouraged to compare with, such as SPPO [1] and RainbowPA [2].**
>
> Thank you for the valuable suggestion, which can further enrich our work. We are also aware of the SSPO and RainbowPA papers. SSPO uses the same evaluation set and metrics as we do, and its code has not been open-sourced. Therefore, for a clear comparison, we directly quote the results reported in the SSPO paper. In addition, the ImageReward scores in the SSPO paper are not directly comparable to other works, likely due to differences in the computation protocol, so we remove this column from our comparison. The resulting table is shown below, where we can see that our SIPO method outperforms SPIN-Diffusion, and SPIN-Diffusion in turn outperforms SSPO.
>
> | Model         | HPS ↑   | Aesthetic ↑  | PickScore ↑ |
> |--------------|---------|-------------|---------------|
> | SD-1.5       | 0.2699  | 5.7691       | 21.1983     |
> | SFT          | 0.2749  | 5.9451       | 21.4542     |
> | Diffusion-DPO| 0.2753  | 5.8918      | 21.8866     |
> | SPIN-Diffusion | 0.2759 | **6.2481**   | 22.0024     |
> | SSPO      | 0.2720  | 5.772     | 21.57   |
> | **SIPO (ours)** | **0.2763** | 6.2451  | **22.0130** |
>
> Similarly, RainbowPA is also trained on SD1.5 with the Pick-a-Pic dataset, but it does not report results on the Pick-a-Pic test set. For a fair comparison, we therefore train SD1.5 on Pick-a-Pic using both RainbowPA and our method, and evaluate the resulting models on GenEval with VQAS, CLIPS, and HPS as metrics. The results are shown in the table below and indicate that our SIPO method achieves comparable performance to RainbowPA on VQAS, while clearly outperforming RainbowPA on both CLIPS and HPS.
> | GenEval        | VQAS ↑ | CLIPS ↑ | HPS ↑ |
> |----------------|--------|---------|-------|
> | SD-1.5         | 61.85  | 33.77   | 26.93 |
> | Diffusion-DPO  | 63.68  | 34.52   | 27.17 |
> | SPIN-Diffusion | 60.81  | 32.79   | 27.31 |
> | SePPO          | 63.10  | 33.94   | 27.35 |
> | RainbowPA       | **68.41** | 35.01 | 27.45 |
> | **SIPO (ours)** | 68.37 | **36.31**  |  **27.52** |
>
> Finally, we would like to reiterate our gratitude to the reviewer. Should further experiments be deemed necessary, we are prepared to carry them out and incorporate the results accordingly.

---

> > ### Author Response · Authors · 2025-11-25
> > **Official comment by Authors**
> >
> > Thank you for your efforts and valuable suggestions that have significantly improved the quality of our paper.
> >
> > We have carefully addressed each of your concerns in our rebuttal and we sincerely hope our rebuttal addressed your concerns. In the revised manuscript, we have updated our results in Table 1 (**page 8, lines 378-391**) to accurately reflect the observations and conclusions of our claims (**page 2, line 101-107**). We have also added a detailed description of our human evaluation protocol (**Sec. E, page 21**) and included a sensitivity analysis on hyperparameters (**Sec. F, page 23**). Furthermore, our *point-by-point* responses provide additional clarification regarding your specific concerns. We sincerely thank you again for your constructive suggestions for improving our presentation.
> >
> > We would be grateful if you could consider providing further feedback if time allows, we are more than happy to answer any further questions you may have!
> >
> > Best regards,
> >
> > Authors.

---

### Official Review · Reviewer_tiFh · 2025-10-31

**Soundness:** 3
**Presentation:** 3
**Contribution:** 3
**Rating:** 6
**Confidence:** 4

**Summary:**

This paper addresses the instability observed in applying Direct Preference Optimization (DPO) to diffusion models. The authors propose two complementary improvements:
(1) DPO-C&M, which introduces timestep-dependent importance masking and gradient clipping to mitigate gradient explosion and overemphasis on uninformative steps; and
(2) SIPO, which modifies the DPO objective by applying clipped importance reweighting to the log-likelihood ratio term and reformulates the loss as KL minimization toward a reward-shaped target distribution. By skipping early diffusion steps and leveraging importance sampling, SIPO aims to improve training stability and convergence behavior.

**Strengths:**

oth modifications (C&M and SIPO) are lightweight yet principled, and can be easily integrated into existing preference-optimized diffusion frameworks.
The experiments demonstrate noticeable stability improvements across image and video generation tasks, suggesting the methods’ robustness in practice.

**Weaknesses:**

1. Inconsistent β values across methods. Table A1 shows that β differs between baselines and proposed methods (e.g., for video: DPO=2 vs. SIPO/DPO-C&M=0.02; for image: DPO=5 vs. SIPO/DPO-C&M=1). Since β directly controls the conservative/aggressive trade-off in DPO, this discrepancy could systematically favor the proposed variants. The authors should present comparisons under matched β settings or include grid-based sensitivity analyses with statistical significance.

2. Lack of human evaluation details. More information about the human annotation setup is needed—number of raters, cleaning or filtering criteria, and aggregation methods. Releasing minimal anonymized examples would further enhance reproducibility and transparency.

**Questions:**

see the weakness

---

> ### Author Response · Authors · 2025-11-21
> **Response to Reviewer tiFh**
>
> >**Weaknesses 1:Inconsistent β values across methods. Table A1 shows that β differs between baselines and proposed methods (e.g., for video: DPO=2 vs. SIPO/DPO-C&M=0.02; for image: DPO=5 vs. SIPO/DPO-C&M=1). Since β directly controls the conservative/aggressive trade-off in DPO, this discrepancy could systematically favor the proposed variants.**
>
> Thank you for the careful review and the very helpful comment. We acknowledge that our explanation of the β setting was not sufficiently clear, and we would like to clarify it here; if this is still unsatisfactory, we are willing to rerun the experiments with matched β and update the paper.
>
> First, on both the video model CogVideoX-2B and the image model SD1.5, we explicitly verified that our method is insensitive to the β hyperparameter (see Fig. 1(a) and Fig. 8(c)): when we sweep β from 0.02 to 5, our method consistently achieves similarly strong performance. This is why we view the exact numerical choice of β for SIPO/DPO-C&M as relatively uncritical.
>
> For ease of review, we convert the corresponding curves into explicit numerical tables: for the CogVideoX-2B video model we report the VBench metric, whereas for the SD1.5 image generation model we report the HPSv2 metric.
>
> | ($\beta$) | CogVideoX-2B (baseline) | Diffusion-DPO | SIPO  |
> | ------- | ----------------------- | ------------- | ----- |
> | 0.01    | 80.91                   | 70.49         | 81.39 |
> | 0.02    | 80.91                   | 73.81         | 81.51 |
> | 1       | 80.91                   | 77.81         | 81.46 |
> | 5       | 80.91                   | 81.14         | 81.49 |
> | 10      | 80.91                   | 81.02         | 81.37 |
>
> |($\beta$）     | SD1.5 (baseline)  | DiffusionDPO | SIPO  |
> |-------|----------|--------------|-------|
> | 0.01  | 26.85    | 15.4         | 26.5  |
> | 0.02  | 26.85    | 18.8         | 27.3  |
> | 1     | 26.85    | 26.9         | 27.5 |
> | 5     | 26.85    | 27.7         | 27.9  |
> | 10    | 26.85    | 26.9         | 27.4  |
>
> Second, our intention was to *stress-test stability*: we deliberately use relatively small β for our methods in the main experiments and still obtain SOTA results, while Diffusion-DPO and other baselines do not perform well under such small β (this behavior is also reported in the original Diffusion-DPO paper). Using the same small β as ours for these baselines simply does not work, whereas increasing β for our method does not hurt its performance. Thus, the β configuration is not chosen to favor our method; on the contrary, our method is evaluated under a more challenging, low-β regime, which empirically supports our claim of more stable training.
>
> >**Weaknesses 2:Lack of human evaluation details. More information about the human annotation setup is needed—number of raters, cleaning or filtering criteria, and aggregation methods. Releasing minimal anonymized examples would further enhance reproducibility and transparency.**
>
> We thank the reviewer for this very helpful suggestion, which indeed helps us improve the clarity and rigor of our work.
>
> **Human evaluation details**   Concretely, we use **200 prompts** covering a broad range of cases (humans, animals, attributes, single/multi-object scenes, food, vehicles, etc.). For each prompt, two models generate outputs under the same sampling settings, and the resulting pair is shown in an A/B side-by-side interface with randomized left/right order and anonymized model identities. We recruit 12 annotators(6 male, 6 female, aged 21–25, all with a bachelor-level education), and split them into 3 groups of 4. Each group independently labels all 200 pairs by choosing which result is better (or “tie”), based on both prompt alignment and visual quality. We first take a majority vote within each group* and then a majority vote across the 3 groups** to obtain the final label per pair; ties are treated accordingly. This two-stage aggregation, together with randomization and anonymization, is designed to ensure a fair and robust human evaluation.
>
> We have incorporated a detailed description of this protocol, together with example prompts and aggregation statistics, into Appendix E of the revised article.
>
> **Examples**   This is a very good suggestion. We have converted the examples into an anonymous link for your reference, which is provided below.
> https://anonymous.4open.science/r/ICLR2026_SIPO_anonymized-58CE ;they are available under the **example_video_case** directory at the following link.

---

> > ### Author Response · Authors · 2025-11-25
> > **Official comment by Authors**
> >
> > Thank you for your efforts and valuable suggestions that have significantly improved the quality of our paper.
> >
> > We have carefully addressed each of your concerns in our rebuttal and we sincerely hope our rebuttal addressed your concerns. In the revised manuscript, we have updated our results in Table 1 (**page 8, lines 378-391**) to accurately reflect the observations and conclusions of our claims (**page 2, line 101-107**). We have also added a detailed description of our human evaluation protocol (**Sec. E, page 21**) and included a sensitivity analysis on hyperparameters (**Sec. F, page 23**). Furthermore, our *point-by-point* responses provide additional clarification regarding your specific concerns. We sincerely thank you again for your constructive suggestions for improving our presentation.
> >
> > We would be grateful if you could consider providing further feedback if time allows, we are more than happy to answer any further questions you may have!
> >
> > Best regards,
> >
> > Authors.

---

### Official Review · Reviewer_gHDb · 2025-11-01

**Soundness:** 3
**Presentation:** 3
**Contribution:** 3
**Rating:** 4
**Confidence:** 4

**Summary:**

The paper proposes SIPO (Stabilized and Improved Preference Optimization) for aligning diffusion models with human (or AI) preferences. The key ideas are: (1) identify that early timesteps in diffusion contribute high-variance, low-importance gradients; (2) introduce DPO-C&M (clipping & masking) using timestep-wise importance weights to stabilize training; and (3) further correct off-policy bias via importance-weighted DPO with clipped, timestep-aware weights. Experiments on SD1.5/SDXL for T2I and CogVideoX/Wan for T2V claim improved stability and accuracy over Diffusion-DPO and other baselines, with lower sensitivity to β and better human evals.

**Strengths:**

1. Clear empirical diagnosis of instability: The paper argues and shows that early timesteps have low importance weights and introduce noisy gradients; masking/clipping these improves stability.
2. Principled off-policy correction: Casting diffusion DPO with importance sampling and clipping is well-motivated by RL literature; the step-wise treatment is natural for diffusion.
3. Breadth of evaluation: Benchmarks across SD1.5/SDXL (T2I) and CogVideoX/Wan (T2V) with automatic metrics and human ranking; reports reduced β-sensitivity and smoother learning curves.

**Weaknesses:**

1. In line 361, you state “at 1000 steps, Diffusion-DPO collapses (67.28)” whereas Table 1 shows pretrained = 67.28 and Diffusion-DPO = 81.46. This contradiction undermines the claimed failure of Diffusion-DPO at longer training. Please reconcile.

2. Missing ablations: Choices like threshold 0.9 for pruning, the clipping range [1−ε,1+ε], and their per-dataset sensitivity lack rigorous ablations; Importance estimation per-timestep may add overhead; wall-clock comparisons to Diffusion-DPO/SPIN etc. are not reported.

**Questions:**

1. Please clarify the results in Table 1.

2. Please add more ablations such as sensitivity and justification for the 0.9 threshold and ε range across datasets.

---

> ### Author Response · Authors · 2025-11-21
> **Response to Reviewer gHDb (Part I)**
>
> >***Weaknesses 1: In line 361, you state “at 1000 steps, Diffusion-DPO collapses (67.28)” whereas Table 1 shows pretrained = 67.28 and Diffusion-DPO = 81.46. This contradiction undermines the claimed failure of Diffusion-DPO at longer training. Please reconcile.**
>
> Thank you for the careful reading and for pointing out this inconsistency. You are correct that the sentence at line 361 and the numbers in Table 1 are contradictory. This discrepancy is due to a late-stage formatting mistake: two rows were accidentally duplicated, which led to the wrong values being shown for the CogVideoX-2B @ 1000 steps setting.The complete and corrected VBench results are as follows:
>
> | Model                        | Method           | Total ↑ | Quality ↑ | Semantic ↑ |
> |-----------------------------|------------------|---------|-----------|------------|
> | **CogVideoX-2B @ 500 Steps**|                  |         |           |            |
> | CogVideoX-2B @ 500 Steps    | Pretrained       | 80.91   | 82.18     | 75.83      |
> | CogVideoX-2B @ 500 Steps    | + Diffusion-DPO  | 81.16   | 82.32     | 76.49      |
> | CogVideoX-2B @ 500 Steps    | + C&M            | 81.37   | 82.55     | 76.69      |
> | CogVideoX-2B @ 500 Steps    | **+ SIPO (ours)**| **81.53** | **82.74** | **76.71**  |
> |                             |                  |         |           |            |
> | **CogVideoX-2B @ 1000 Steps**|                 |         |           |            |
> | CogVideoX-2B @ 1000 Steps   | Pretrained       | 80.91   | 82.18     | 75.83      |
> | CogVideoX-2B @ 1000 Steps   | + Diffusion-DPO  | 67.28   | 68.37     | 62.93      |
> | CogVideoX-2B @ 1000 Steps   | + C&M            | 81.46   | 82.65     | 76.72      |
> | CogVideoX-2B @ 1000 Steps   | **+ SIPO (ours)**| **81.68** | **82.92** | **76.76**  |
>
> With the corrected table, our original claim is consistent: for CogVideoX-2B at 1000 steps, Diffusion-DPO indeed collapses (Total score drops from 80.91 to 67.28), while both C&M and SIPO remain stable and even slightly improve over the pretrained model. We will fix the typo and update the table accordingly in the revised version. We sincerely apologize for the confusion caused.

---

> > ### Author Response · Authors · 2025-11-21
> > **Response to Reviewer gHDb (Part II)**
> >
> > >**Weaknesses 2: Missing ablations: Choices like threshold 0.9 for pruning, the clipping range [1−ε,1+ε], and their per-dataset sensitivity lack rigorous ablations; Importance estimation per-timestep may add overhead; wall-clock comparisons to Diffusion-DPO/SPIN etc. are not reported.**
> >
> > **Sensitivity of the pruning threshold and clipping range.**
> >
> > Thank you for raising this point. Our choice of the pruning threshold is mainly motivated by two considerations. First, as visualized in Fig. 4, when the importance weight threshold is set to 0.9, roughly the first ~60 timesteps are pruned, and our experiments show that discarding these low-signal, high-variance timesteps consistently stabilizes DPO training. Second, this choice is in line with common practice in PPO/GRPO, where clipping ratios of the form ([1-$\varepsilon$, 1+$\varepsilon$]) with ($\varepsilon$ in {0.1, 0.2}) are widely used as heuristic defaults; we similarly treat the pruning threshold as a robust heuristic rather than a finely tuned hyperparameter.
> >
> > To directly address the sensitivity of ($\varepsilon$), we additionally trained SD1.5 on the Pick-a-Pic v1 dataset (evaluated on the Pick-a-Pic test set) and SDXL on the Pick-a-Pic v2 dataset (evaluated on the HPSv2 test set to avoid contamination of the Pick-a-Pic test set), using HPSv2 score as the evaluation metric. We swept ($\varepsilon$ in {0.05, 0.1, 0.2, 0.3, 0.4, 0.5); the results (“The results are summarized in the table below, and we will include clearer curve plots in the revised manuscript.) indicate that ($\varepsilon$ = 0.1) achieves the highest score, with ($\varepsilon$ = 0.2) very close. When ($\varepsilon$ = 0.05), too many timesteps are effectively clipped, leading to under-training and worse performance. As ($\varepsilon$) increases beyond 0.2, relaxing the constraint allows larger importance ratios, which increases the variance of the gradient estimates and causes a gradual performance drop, consistent with standard importance sampling theory. Importantly, SD1.5 and SDXL exhibit the same qualitative trend across ($\varepsilon$), suggesting that SIPO is not overly sensitive to the specific dataset or backbone.
> >
> > | Model | Baseline |  0.05  |  0.10  |  0.20  |  0.30  |  0.40  |  0.50  |
> > |:-----:|:--------:|:------:|:------:|:------:|:------:|:------:|:------:|
> > | SDXL  | 0.2812   | 0.2853 | 0.2889 | 0.2885 | 0.2842 | 0.2833 | 0.2825 |
> > | SD1.5 | 0.2699   | 0.2720 | 0.2763 | 0.2759 | 0.2735 | 0.2718 | 0.2691 |
> >
> >
> > **Computational overhead.**
> >
> > SPIN-Diffusion does not have an official public implementation, so we are unable to report a fair wall-clock comparison for that method. Compared to Diffusion-DPO, however, SIPO does not introduce any additional trainable parameters. In Diffusion-DPO training we already obtain both ($q(x_{t-1}\mid x_t)$) and ($p(x_{t-1}\mid x_t)$); SIPO simply plugs these into the closed-form Gaussian densities to compute the importance weights. This computation is done without gradients and without extra network calls, and the cost of evaluating Gaussian terms is negligible relative to the U-Net and reward model forward passes. Therefore, in practice SIPO incurs essentially no additional wall-clock overhead compared to Diffusion-DPO.
> >
> > Thanks again for your efforts and valuable insights, please let us know if you have any further questions.

---

> > > ### Author Response · Authors · 2025-11-25
> > > **Official comment by Authors**
> > >
> > > Thank you for your efforts and valuable suggestions that have significantly improved the quality of our paper.
> > >
> > > We have carefully addressed each of your concerns in our rebuttal and we sincerely hope our rebuttal addressed your concerns. In the revised manuscript, we have updated our results in Table 1 (**page 8, lines 378-391**) to accurately reflect the observations and conclusions of our claims (**page 2, line 101-107**). We have also added a detailed description of our human evaluation protocol (**Sec. E, page 21**) and included a sensitivity analysis on hyperparameters (**Sec. F, page 23**). Furthermore, our *point-by-point* responses provide additional clarification regarding your specific concerns. We sincerely thank you again for your constructive suggestions for improving our presentation.
> > >
> > > We would be grateful if you could consider providing further feedback if time allows, we are more than happy to answer any further questions you may have!
> > >
> > > Best regards,
> > >
> > > Authors.

---

### Author Response · Authors · 2025-12-02
**Brief summarization of rebuttal and revisions**

**Brief summarization of rebuttal and revisions**

We sincerely thank all reviewers for their careful assessments and constructive suggestions. Specifically, we are encouraged that our method was acknowledged as "**Clear empirical diagnosis of instability**" and "**Principled and well-motivated off-policy correction**" by Reviewer `gHDb`, "**can be easily integrated into existing preference-optimized diffusion frameworks**" by Reviewer `tiFh`, "**more stable training**" and "**shows remarkable robustness**" by Reviewer `X295`. Moreover, our experimental results were praised as **extensive and high-quality**, covering diverse image and video models (SD1.5/SDXL, CogVideoX, Wan, FLUX-dev) with both automatic metrics and human evaluations, and consistently showing **improved stability, robustness, and reduced β-sensitivity** compared to Diffusion-DPO.

Our rebuttal mainly addresses below concerns: 1) fix table-formatting glitches; 2) add and clarify ablations on key hyperparameters ($\beta$, $\epsilon$)—where ablations on $\beta$ were already presented in the original paper, and the choice of $\epsilon$ is now added by detailed curve analysis and additional experiments on different $\epsilon$; and 3) clarify computational cost and providing more details of the human evaluation protocol.

1. **Clarification on hyperparameter choices ($\beta$, $\epsilon$).**
   We now emphasize that we already perform systematic ablations on the DPO coefficient $\beta$ for both image (SD1.5) and video (CogVideoX-2B) models in the paper. These experiments show that our SIPO method is robust to $\beta$, whereas vanilla Diffusion-DPO requires a relatively large $\beta$ to maintain reasonable performance. To highlight stability, in the main experiments we therefore intentionally use a smaller $\beta$ for SIPO and a larger $\beta$ for Diffusion-DPO.

   For the pruning/clipping threshold $\epsilon$, we describe in more detail how we first select $\epsilon$ based on performance curves, and we have added experiments sweeping $\epsilon$ over a wide range. The results show that SIPO is not sensitive to $\epsilon$ in a reasonable interval and validate the choice used in the main text.

2. **Correction of VBench results .**
   We discovered a late-stage formatting mistake in the CogVideoX-2B 1000-step setting that duplicated two rows and led to inconsistent VBench scores in Table 1. We have corrected these results and provided the full table in the rebuttal. After correction, Diffusion-DPO clearly collapses at 1000 steps (its total VBench score drops significantly below the pretrained model), while both C&M and SIPO remain stable or slightly improve, supporting our claim about long-horizon instability of Diffusion-DPO.

3. **Analysis of timestep ranges and instability.**
   We clarify our earlier statement about “early and late timesteps” and add new experiments where SD1.5 is trained with different timestep ranges (only early, middle-to-late, and full range). The results show that removing very early timesteps brings the main improvement, whereas additionally removing very late timesteps has only minor impact. We accordingly revise the introduction to emphasize that instability primarily arises from noisy early timesteps.

4. **Computational cost and human evaluation protocol.**
   We further explain that SIPO introduces no extra trainable parameters and reuses quantities already computed in Diffusion-DPO training; the additional Gaussian computations are negligible compared to U-Net forward passes, so SIPO brings virtually no extra wall-clock cost.
In response to requests for more details on human evaluation, we substantially expand the description of our protocol: the construction of 200 diverse prompts, the A/B side-by-side interface with randomized left/right order and anonymized model identities, the recruitment and grouping of 12 annotators, and the two-stage majority-vote aggregation. We also provide anonymized example pairs to facilitate reproduction.

5. **Additional baselines and fair comparison.**
   The methods suggested by the reviewers (such as SSPO and RainbowPA) are generally somewhat weaker than the strongest baselines we already compare to, which is why we originally omitted them from the main tables. We agree that including them improves completeness and fairness. Therefore, we have added SSPO and RainbowPA: for SSPO, we report the official results under the same evaluation protocol; for RainbowPA, we implement it on SD1.5 and evaluate it on GenEval (VQAS, CLIPS, HPS). The updated tables show that SIPO remains competitive and often superior to these methods while also outperforming or matching other SoTA baselines.

Overall, these revisions strengthen our main conclusion: SIPO offers a simple and computationally efficient way to stabilize preference-based diffusion training, exhibiting robustness to hyperparameters and achieving strong performance across image and video generation benchmarks.

---

### Meta-Review · Area_Chair_o65U · 2026-01-06

**Summary:**

The paper analyzes the sources of instability in diffusion‑based preference learning and introduces SIPO, a Stabilized and Improved Preference Optimization framework for aligning diffusion models with human preferences. Reviewers’ concerns about missing ablations, claim contradictions, and related issues have been mostly addressed. However, the response to one major concern regarding the lack of analysis on late‑stage instability could be further clarified.

**Reviewer Concerns:**

Some of the reviewers’ concerns have been addressed. The authors added more results and discussions regarding choices such as the pruning threshold of 0.9 and the clipping range. Although SIPO appears to incur no significant overhead compared to Diffusion‑DPO, a wall‑clock comparison would still be helpful, as requested by one reviewer. The authors also included results across various β values. One major concern, however, involves the missing analysis of “why the very late timesteps might also be unstable or uninformative, and how SIPO’s mechanisms specifically address this tail‑end instability,” while the core analysis focuses heavily on early timesteps being problematic due to low importance weights. The authors responded by stating that “the statement that ‘early and late stages introduce instability’ at the end of the introduction is a writing mistake.” The AC finds this concern valid and important, and the authors’ response could be strengthened, for example, by more clearly explaining the claimed failure of Diffusion‑DPO during longer training, if not from instability, and how SIPO helps.

**Reviewer Scores:**

This paper receives the following ratings: Marginally Below, Marginally Above, and Marginally Below. If the reviewers had been able to participate fully in the discussion, the AC would expect at least one negative rating to still remain. The AC recommends not accepting the paper in its present form.

---

### Decision · Program_Chairs · 2026-01-26

Reject